# Ring finger protein 213 assembles into a sensor for ISGylated proteins with antimicrobial activity

Fabien Thery[1,2,14], Lia Martina[1,2,14], Caroline Asselman [1,2,14], Yifeng Zhang [3], Madeleine Vessely[3], Heidi Repo[1,2], Koen Sedeyn [1,4], George D. Moschonas [1,4], Clara Bredow [5], Qi Wen Teo[6], Jingshu Zhang[6], Kevin Leandro[1,2], Denzel Eggermont [1,2], Delphine De Sutter[1,2], Katie Boucher[1,2,7], Tino Hochepied[8,9], Nele Festjens[1,4], Nico Callewaert [1,4], Xavier Saelens [1,4], Bart Dermaut [2,10], Klaus-Peter Knobeloch [11], Antje Beling [5,12], Sumana Sanyal [6,13], Lilliana Radoshevich[3✉], Sven Eyckerman [1,2✉] & Francis Impens [1,2,7✉]

ISG15 is an interferon-stimulated, ubiquitin-like protein that can conjugate to substrate proteins (ISGylation) to counteract microbial infection, but the underlying mechanisms remain elusive. Here, we use a virus-like particle trapping technology to identify ISG15-binding proteins and discover Ring Finger Protein 213 (RNF213) as an ISG15 interactor and cellular sensor of ISGylated proteins. RNF213 is a poorly characterized, interferon-induced megaprotein that is frequently mutated in Moyamoya disease, a rare cerebrovascular disorder. We report that interferon induces ISGylation and oligomerization of RNF213 on lipid droplets, where it acts as a sensor for ISGylated proteins. We show that RNF213 has broad antimicrobial activity in vitro and in vivo, counteracting infection with *Listeria monocytogenes*, herpes simplex virus 1, human respiratory syncytial virus and coxsackievirus B3, and we observe a striking co-localization of RNF213 with intracellular bacteria. Together, our findings provide molecular insights into the ISGylation pathway and reveal RNF213 as a key antimicrobial effector.

[1] VIB-UGent Center for Medical Biotechnology, VIB, Ghent, Belgium. [2] Department of Biomolecular Medicine, Ghent University, Ghent, Belgium. [3] Department of Microbiology and Immunology, University of Iowa Carver College of Medicine, Iowa City, IA, USA. [4] Department of Biochemistry and Microbiology, Ghent University, Ghent, Belgium. [5] Charité—Universitätsmedizin Berlin, corporate member of Freie Universität Berlin and Humboldt-Universität zu Berlin, Institute of Biochemistry, Berlin, Germany. [6] HKU-Pasteur Research Pole, School of Public Health, University of Hong Kong, Pok Fu Lam, Hong Kong. [7] VIB Proteomics Core, VIB, Ghent, Belgium. [8] VIB Center for Inflammation Research, VIB, Ghent, Belgium. [9] Department of Biomedical Molecular Biology, Ghent University, Ghent, Belgium. [10] Center for Medical Genetics, Ghent University Hospital, Ghent, Belgium. [11] Institute of Neuropathology, Medical Faculty, University of Freiburg, Freiburg, Germany. [12] Deutsches Zentrum für Herz-Kreislauf-Forschung (DZHK), partner side Berlin, Berlin, Germany. [13] Sir William Dunn School of Pathology, University of Oxford, Oxford OX1 3RE, UK. [14] These authors contributed equally: Fabien Thery, Lia Martina, Caroline Asselman. ✉email: lilliana-radoshevich@uiowa.edu; sven.eyckerman@vib-ugent.be; francis.impens@vib-ugent.be

SG15 is a ubiquitin-like (UBL) protein with antimicrobial activity. Similar to ubiquitin, ISG15 conjugates via its C-terminus to lysine residues of substrate proteins in a process called ISGylation, which is mediated by an E1 enzyme, UBE1L, an E2 enzyme, UBCH8, and three known E3 enzymes, HHARI, TRIM25, and HERC5[1–5]. ISG15 and its conjugation machinery are strongly upregulated by Type I and III interferon, viral nucleic acids[1], bacterial DNA[6], and lipopolysaccharide (LPS)[7]. ISG15 has potent antiviral effects both in vitro and in vivo[8]. In fact, cells or mice which lack ISG15 are unable to control various viral pathogens including clinically relevant etiologic agents such as Influenza[9], human respiratory syncytial virus[10], and coxsackievirus[11,12]. ISG15's crucial antiviral role is bolstered by the variety of viral evasion strategies targeting the ISGylation pathway, either by interfering with ISG15 conjugation[1,13] or by expressing ISG15-specific proteases that lead to reversible[14–17] or irreversible deconjugation[18]. In addition to its antiviral role, ISG15 can also act as an antibacterial effector against intracellular bacterial pathogens such as *Listeria monocytogenes*[6] and *Mycobacterium tuberculosis*[19] and it can restrict the intracellular eukaryotic pathogen *Toxoplasma gondii*[20]. Despite its broad antimicrobial role, the molecular mechanisms by which ISG15 modification is sensed and how it protects against microbial infections remain largely undefined. One model posits that the antiviral function of ISG15 is based on the localization of the E3 ligase HERC5 at the ribosome. According to this model, proteins are co-translationally modified by ISG15 during infection, thereby interfering with the function of newly translated viral proteins in a nonspecific manner[21]. However, the model does not predict what subsequently happens to ISGylated proteins in the cell and how they are recognized and trafficked.

Unlike ubiquitin, ISG15 does not solely exert its antimicrobial function by covalently conjugating to target proteins. It can also be secreted and act as a cytokine[22–24] and can noncovalently interact with viral and host proteins[25]. Relatively little is known about what interaction partners bind ISG15 and how the functions of these interactions contribute to host responses to pathogens. Indeed, only a few host proteins have been reported to noncovalently bind to ISG15, mainly in its free form. One of the best-characterized ISG15-binding proteins is USP18, the predominant ISG15 deconjugating protease in human and mice[26,27]. In addition to its enzymatic activity, USP18 functions as a major negative regulator of type I interferon (IFN-I) signaling by binding to one of the subunits of the interferon α/β receptor (IFNAR2)[28–30]. In order to prevent IFN-I over-amplification in humans, USP18 needs to be stabilized by binding to free ISG15 in a noncovalent manner. Hence, patients who lack ISG15 expression due to a frameshift mutation have a strong upregulation of the IFN-I pathway leading to an Aicardi-Goutières-like interferonopathy[31]. In contrast to *Isg15*-deficient mice, these patients do not display enhanced susceptibility to viral infection, instead they are susceptible to bacterial infection including clearance of the attenuated tuberculosis vaccine, Bacille Calmette-Guerin (BCG)[32]. In addition to USP18, a few other proteins have been reported to interact with ISG15 in a noncovalent manner. Free intracellular ISG15 can bind to leucine-rich repeat-containing protein 25 (LRRC25) and mediate the autophagic degradation of retinoic acid-inducible gene I protein (RIG-I/DDX58)[33]. Upon forced overexpression, p62 and HDAC6 were further shown to interact with the C-terminal LRLRGG sequence of ISG15 (and ISGylated proteins)[34]. Free ISG15 also interacts with the E3 ligase NEDD4 and ISG15 overexpression disrupts NEDD4 ubiquitination activity thus blocking the budding of Ebola virus-like particles[35,36]. Likewise, free ISG15 was reported to interact in a noncovalent manner with the Hypoxia inducible factor 1α (HIF1α), preventing its dimerization and downstream signaling[37].

ISG15 has not been studied as extensively as ubiquitin or SUMO for which interacting domains or motifs have been reported, thus to date no such domains or motifs have been described for ISG15. Although affinity purification mass spectrometry (AP-MS) and yeast two-hybrid (Y2H) screens were performed with ISG15 as bait, these interactome screens were hampered by the inherently limited binding surface of such a small protein and the stringency of yeast two-hybrid, which would preclude low affinity interactions[38–41]. Therefore, we endeavored to identify noncovalent interaction partners of ISG15 using a recently developed approach named Virotrap, which is based on capturing protein complexes within virus-like particles (VLPs) that bud from mammalian cells[42]. Virotrap is a cell-lysis-free method and thus preserves existing protein complexes, and, importantly, this method is uniquely suited to capture weak and transient cytosolic interactions that otherwise do not survive protein complex purification. Using this technology, we systematically mapped the noncovalent interactome of ISG15 in human cells. We identified a rich interactome but for further study focused on a very large protein, called Ring Finger Protein 213 (RNF213), mutations in which predisposes patients to early cerebrovascular events. Subsequent biochemical evaluation confirmed the selective binding of RNF213 to ISG15 and showed that IFN-I signaling induces RNF213 oligomerization into a sensor for ISGylated proteins associated with lipid droplets. Finally, in vitro and in vivo loss and gain of function experiments showed that RNF213 is a pivotal, broadly acting microbial restriction factor that colocalizes to the surface of intracellular bacteria.

## Results

**Identification of RNF213 as an ISG15-binding protein by Virotrap**. To investigate the noncovalent interactome of ISG15 and cytosolic sensors for ISG15 modification, we used Virotrap, a recently developed MS-based approach to map protein–protein interactions[42]. Briefly, the sequence of mature ISG15 (ending on -LRLRGG) was genetically fused via its N-terminus to the GAG protein of HIV-1 and expressed in HEK293T cells treated with IFN-I. Expression of the GAG-ISG15 fusion protein led to self-assembly of VLPs internally coated with ISG15. After budding from the cells and capturing ISG15 interaction partners, VLPs were collected from the cell culture supernatants and used for MS-based protein identification (Fig. 1a). To distinguish specific ISG15-binding proteins from nonspecific (GAG-binding) proteins, the experiment was performed in a quantitative fashion comparing ISG15 VLPs with VLPs containing dihydrofolate reductase from *Escherichia coli* (eDHFR), a nonspecific bait protein with no obvious interaction partners in human cells. In addition to ISG15, we identified RNF213 as the most enriched protein in the ISG15 VLPs, associated with high spectral and peptide counts (Fig. 1b, Supplementary Data 1). Of note, among several less enriched proteins we also identified USP18, which is an ISG15-specific isopeptidase and thus a well-known interaction partner of ISG15[26,27]. To rule out any type of covalent interaction between GAG-ISG15 and RNF213, we repeated the Virotrap experiment using the nonconjugatable ISG15AA (with the c-terminal glycines mutated to alanines: LRLRAA) and precursor variants of ISG15, both with and without IFN-I treatment (Supplementary Fig. 1, Supplementary Data 1). Without exception, in all conditions we identified RNF213 as the most enriched protein in the ISG15 VLPs. We also confirmed the presence of RNF213 in the ISG15 VLPs by western blotting (Fig. 1c).

In addition to RNF213, 28 other proteins were enriched in ISG15 VLPs under all conditions (Supplementary Fig. 1, Supplementary Data 1). Following RNF213, the most notable among these identifications was the valosin-containing protein (VCP/p97).

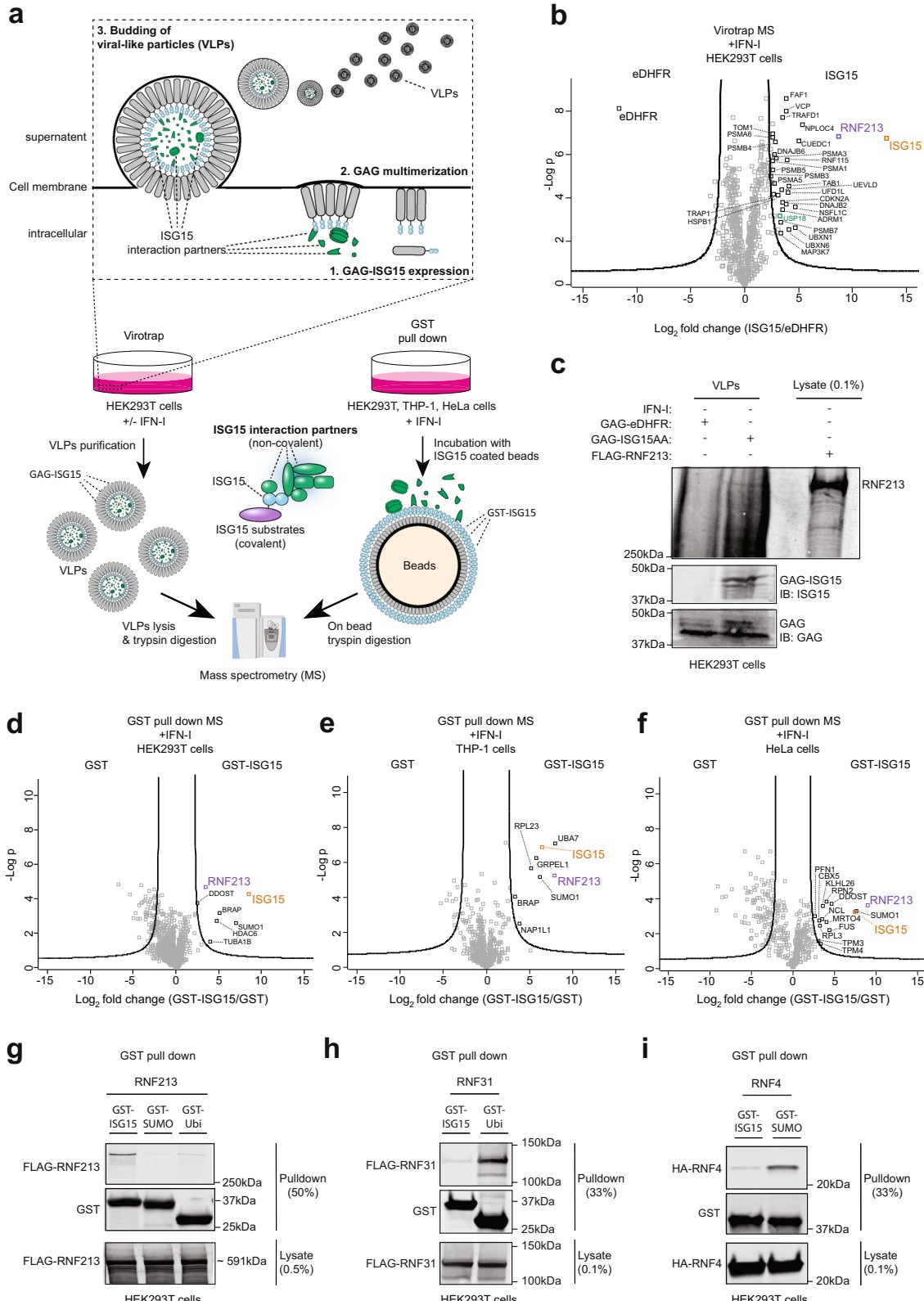

VCP is a hexameric AAA + ATPase with a central role in the endoplasmic-reticulum-associated protein degradation (ERAD) pathway, where it associates with cofactor proteins to facilitate the proteasomal degradation of misfolded proteins[43]. In addition, several cofactors of VCP were found to interact with ISG15 such as the VCP-UFD1-NPL4 protein complex as well as PLAA, UBXN6 and YOD1[44], FAF1, NSFL1C, UBXN1, and RNF115[45–48]. We also

detected several components of the proteasome complex enriched in ISG15 VLPs, including ADRM1 and several PSMA and PSMB subunits of the proteasome. Moreover, we identified TAB1 and TAK1 (MAP3K7), which are involved in innate immune signaling pathways as well as TRAFD1[49,50].

We then performed an additional series of Virotrap-MS experiments in which only the N- or C-terminal domain of

**Fig. 1 Identification of RNF213 as ISG15-binding protein. a** Workflow to map ISG15 interaction partners. Virotrap and Glutathione-S-Transferase (GST) pull-down were employed as two orthogonal methods. In Virotrap, GAG-ISG15 was expressed in HEK293T cells. Twenty four hours post-transfection, cells were treated with interferon (IFN)-α for 24 h of left untreated. Budding virus-like particles (VLPs) containing ISG15 and its interaction partners were purified, lysed and digested into peptides prior to LC-MS/MS analysis. In GST pull-down, glutathione beads were decorated with GST-ISG15 and mixed with a cellular lysate from HEK293T, HeLa, or THP-1 cells. Prior to lysis cells were treated for 24 h with interferon-α (HEK293T) or interferon-β (HeLa and THP-1). Following on-bead digestion, the resulting peptides were analyzed by LC-MS/MS. **b** Volcano plot showing the result of a *t*-test to compare VLPs containing mature ISG15 versus VLPs containing dihydrofolate reductase from *Escherichia coli* (eDHFR) as negative control (*n* = 4 replicates). Proteins outside the curved lines represent specific ISG15 interaction partners. Proteins identified as ISG15 interaction partners in all virotrap screens are annotated (*n* = 29) and listed in Supplementary Data 1. **c** VLPs containing ISG15 or eDHFR were collected and analyzed by immunoblot (IB) against RNF213, ISG15, and GAG. FLAG-RNF213 purified from a lysate of HEK293T cells was loaded as a positive control and confirmed the presence of an RNF213 band in the ISG15 VLPs. **d–f** Volcano plots comparing GST pulldowns using GST-ISG15-coated beads with GST-coated beads as control in lysates of HEK293T, THP-1, and HeLa cells (*n* = 3). Proteins significantly enriched in PDs with ISG15-coated beads are annotated. **g** GST PDs with ISG15, ubiquitin, and SUMO followed by immunoblotting show binding of RNF213 to ISG15, but not to ubiquitin or SUMO. Beads coated with each UBL were mixed with a lysate of HEK293T cells expressing FLAG-RNF213 and bound proteins were analyzed by immunoblot against FLAG and GST. **h, i** Validation of GST PD assays with ubiquitin and SUMO using RNF31 and RNF4 as known binders of these UBLs, respectively. Assays were performed as in **g**. Source data are provided as a Source Data file.

ISG15 was fused to GAG, alongside with full-length ISG15 and eDHFR as positive and negative controls, respectively. None of the VLPs with the N- or C-terminal ubiquitin-like domain of ISG15 could enrich RNF213, suggesting that both domains are required for the interaction with RNF213 (Supplementary Fig. 2a-c). Like for RNF213, the majority of the 28 aforementioned ISG15 interaction partners also required both domains to interact (Supplementary Fig. 2d), suggesting that these proteins might interact indirectly with ISG15 through binding to RNF213. Taken together, our Virotrap screens identified RNF213 as a robust noncovalent interaction partner of ISG15 along with a number of VCP and proteasome-associated proteins, all requiring both the N- and C-terminal domains of ISG15 to interact.

**RNF213 binds ISG15 but not ubiquitin or SUMO.** To confirm the interaction and specificity between ISG15 and RNF213 through an alternative approach, we first carried out classic immunoprecipitation-MS experiments in IFN-I treated HEK293T and HeLa cells expressing HA-tagged-ISG15AA. In line with previous reports[38–40], HA pull-down of ISG15 did not reveal RNF213 nor any other significant interaction partners (Supplementary Fig. 3a-c). This suggested that affinity between a single ISG15 molecule and RNF213 is too weak to detect, similar to a ubiquitin or SUMO molecule to their respective binding proteins[51,52]. Hence, we applied a Glutathione-S-Transferase (GST) pull-down assay in which we coated glutathione beads with GST-ISG15 and then used these ISG15-decorated beads to search for ISG15-binding proteins in a lysate of HEK293T, HeLa, and THP-1 cells treated with IFN-I. GST-coated beads were used as control and interacting proteins were identified by mass spectrometry. Interestingly, this approach yielded RNF213 as a strongly enriched partner of ISG15 along with several other interactors (Fig. 1d–f); however, only RNF213 was identified in both the GST enrichment and the Virotrap experiments (Supplementary Fig. 3d). Among the other interactors, we detected UBE1L (UBA7) and HDAC6, which are known ISG15 binders in THP-1 and HEK293T cells, respectively[1,34]. To further confirm the specificity of the ISG15/RNF213 interaction, we next combined the pull-down of GST-ISG15 with western blotting. In these experiments, a lysate of HEK293T cells expressing FLAG-RNF213 was mixed with glutathione beads bound to GST-ISG15, GST-SUMO, or GST-ubiquitin. While RNF213 clearly binds to GST-ISG15, only background binding could be observed to GST-SUMO and GST-ubiquitin (Fig. 1g). As a positive control, we confirmed binding of RNF4 and RNF31, known binding partners of GST-SUMO and GST-ubiquitin, respectively[51,53] (Fig. 1h, i).

Together, these data show that RNF213 specifically binds ISG15, rather than ubiquitin or SUMO.

**RNF213 binds ISGylated proteins on lipid droplets.** RNF213, also known as mysterin, is a multidomain megaprotein of unknown function. Polymorphisms in *RNF213* predispose human patients to Moyamoya disease (MMD), a rare vascular brain disease leading to stroke, but the underlying molecular mechanisms of this disease remain elusive[54]. The most abundant RNF213 isoform is 5207 amino acids long and is comprised of two adjacent AAA + ATPase modules with a RING domain[55]. RNF213 is expressed in most cells and tissues[56], but levels can be upregulated by immune and inflammatory signaling[55]. AAA + proteins typically assemble into hexameric toroidal complexes that generate mechanical force through ATP binding/hydrolysis cycles. In the case of RNF213, complex formation is dynamic and driven by ATP binding while ATP hydrolysis mediates disassembly[55]. Interestingly, a recent microscopy study showed colocalization of RNF213 and lipid droplets (LDs), suggesting a model in which monomeric RNF213 is directly recruited from the cytosol and ATP binding drives its oligomerization on the surface of lipid droplets[57]. This model, together with the observation that ISG15 binds RNF213 in Virotrap and GST pull-down experiments— which both present multiple ISG15 bait molecules in close proximity on a concave or convex surface, respectively (Fig. 1a)— led us to hypothesize that RNF213 oligomerizes on LDs into a binding platform for ISG15 and potentially multiply ISGylated proteins.

To test this hypothesis, we first verified whether RNF213 can bind ISGylated proteins. To this end, we pulled down FLAG-RNF213 from lysates of HEK293T cells expressing HA-tagged ISG15 or ISG15AA and its conjugation machinery to induce ISGylation. Western blotting against the HA-tag revealed a smear of ISGylated proteins after FLAG-RNF213 pull-down, but not with FLAG-eGFP pull-down or when ISG15AA was expressed (Fig. 2a). We repeated the experiment with eGFP-tagged RNF213 (Supplementary Fig. 4a) and in both cases did not observe any free ISG15 or ISG15AA associated with RNF213, indicating that in cells RNF213 indeed preferentially senses and binds ISGylated proteins rather than free ISG15. In order to determine whether the ISGylated proteins detected were instead partially degraded RNF213 we included a 1% SDS wash, which reduced the ISGylated proteins (Supplementary Fig. 4b) and even demonstrated that RNF213 could bind ISGylated proteins in trans, which originated from a different cell lysate (Supplementary Fig. 4c, d). These data indicate that RNF213 does bind ISGylated proteins. Finally, we also showed that this binding was

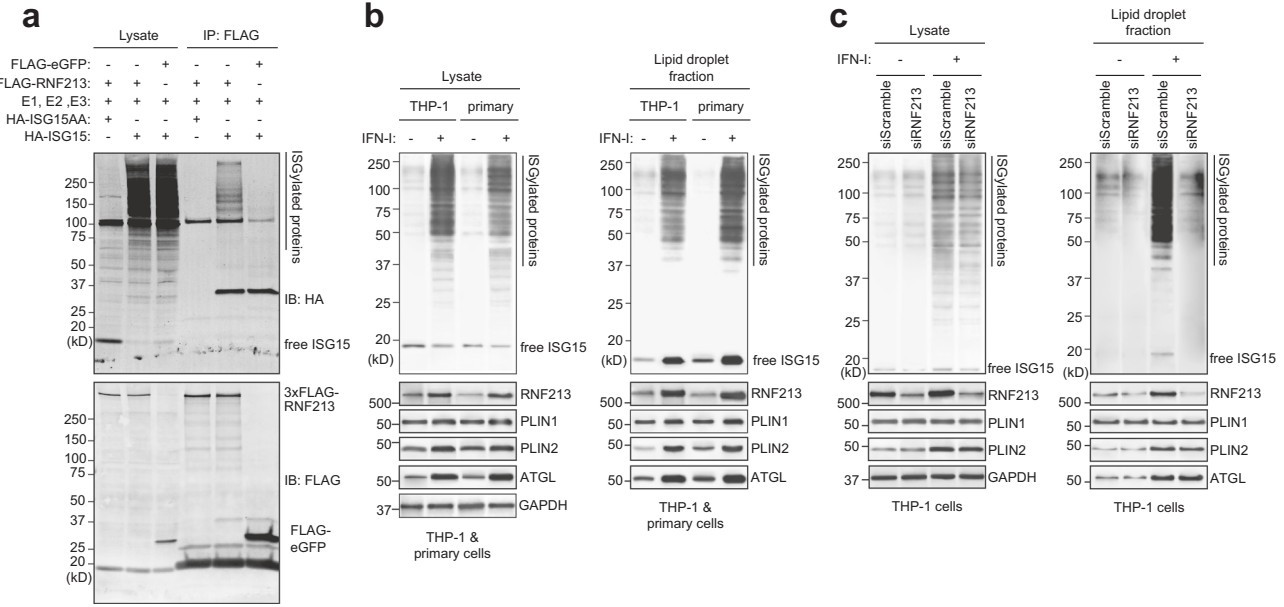

**Fig. 2 RNF213 binds ISGylated proteins on lipid droplets. a** FLAG immunoprecipitation (IP) was performed from lysates of HEK293T cells expressing FLAG-RNF213 or FLAG-eGFP in combination with HA-ISG15(AA) and the ISGylation machinery (E1, E2, E3). A smear of ISGylated co-immunoprecipitated proteins was detected with FLAG-RNF213, but not with FLAG-eGFP or when nonconjugatable HA-ISG15AA was used. **b**. THP-1 or primary human monocytes (CD14+) cells were cultured in the presence of 10 mM BSA-conjugated oleic acid and either treated with 10 ng/mL interferon (IFN) -β for 8 h or left untreated. Lipid droplets (LDs)-enriched fractions were isolated by ultracentrifugation floatation assay on a sucrose step-gradient. Immunoblot (IB) against RNF213 and ISG15 revealed an interferon-induced upregulation of both proteins on LDs and a smear of ISGylated proteins associated with LDs. Immunoblots against PLIN1, PLIN2, ATGL, and GAPDH confirmed LD isolation and equal protein loading in the lysate and LD-enriched fraction (1/20th of the lysate and all of the LD-enriched material was loaded). **c** Similarly, LDs were isolated from THP-1 cells after knockdown of RNF213 by siRNA (siRNF213) treatment for 48 h or using a nontargeting scrambled siRNA (siScramble) as control. Immunoblotting against ISG15 revealed a smear of ISGylated proteins associated with LDs only when RNF213 was present. Immunoblotting against RNF213 confirmed knockdown of RNF213, while PLIN1, PLIN2, ATGL, and GAPDH validated LD isolation and equal protein loading in the lysate and the LD-enriched fraction (1/20th of the lysate and all of the LD-enriched material was loaded). Source data are provided as a Source Data file.

ISG15 specific, since ubiquitinated proteins were not enriched (Supplementary Fig. 4e), in line with the results from the GST pull-down (Fig. 1g).

We next tested whether the interaction between RNF213 and ISGylated proteins occurs on LDs. In HeLa cells treated with IFN-I we observed occasional colocalization of overexpressed eGFP-RNF213 with endogenous ISG15 by fluorescence microscopy; however, ubiquitous intracellular distribution of free ISG15 prevented reliable quantitation of these events. We therefore verified this question further biochemically using macrophages, a cell type which is more relevant for innate immune signaling. We isolated LDs from THP-1 cells and primary human monocytes by flotation on a sucrose gradient[58]. Western blotting for RNF213 and LD marker proteins such as PLIN1 and PLIN2 confirmed that in THP-1 cells the majority of RNF213 is associated with LDs, while a minor fraction is in the cytosol (Supplementary Fig. 5a). Upon IFN-I treatment, the fraction of RNF213 associated with LDs further increased, along with a striking appearance of a smear of ISGylated proteins, an observation that was corroborated in primary human monocytes (Fig. 2b). Finally, to show that ISGylated proteins associate with LDs via RNF213, we repeated the experiment in THP-1 cells in which we reduced the expression of RNF213 by siRNA. As expected, knockdown of RNF213 markedly reduced association of ISGylated proteins with LDs and this occurred without any apparent loss of lipid droplet stability as indicated by unchanged levels of LD markers PLIN1, PLIN2, and ATGL (Fig. 2c). Furthermore, in bone-marrow-derived macrophages (BMDMs) isolated from RNF213 knockout mice[59] we found that the number and size of lipid droplets was

similar to WT BMDMs (Supplementary Fig. 5b, c), indicating that in macrophages RNF213 is not essential for lipid droplet stability, in contrast to other cell types[57]. Together, these data show that RNF213 binds ISGylated proteins and that RNF213 recruits ISGylated proteins to LDs.

**IFN-I induces RNF213 ISGylation and oligomerization on lipid droplets.** We subsequently investigated how oligomerization of RNF213 on LDs is regulated. Since both ISG15 and RNF213 are IFN-I induced genes (Supplementary Fig. 6a)[60], it seemed plausible that multimerization of RNF213 into a sensing platform for ISGylated proteins could be mediated by interferon signaling. We therefore evaluated the effect of IFN-I on the distribution of endogenous RNF213 between the soluble and membrane-associated fractions in HeLa cells and found that the level of membrane-associated RNF213 slightly increased upon IFN-I treatment (Supplementary Fig. 6b). To further confirm RNF213 oligomerization in a relevant model, we separated lysates from control and IFN-I treated THP-1 cells by sedimentation over continuous glycerol gradients. Western blotting of the collected fractions clearly showed that IFN-I treatment gave rise to a pool of higher molecular weight RNF213 in fractions 14–20, as expected after oligomerization (Fig. 3a). Interestingly, this pool of slow migrating RNF213 was marked by the presence of higher molecular weight bands on the blots, most likely representing modified forms of RNF213. Since we recently identified RNF213 as the most ISGylated protein following *Listeria monocytogenes* infection with 22 ISGylation sites[61], we tested whether oligomeric RNF213 was ISGylated. Immunoprecipitating RNF213 from each

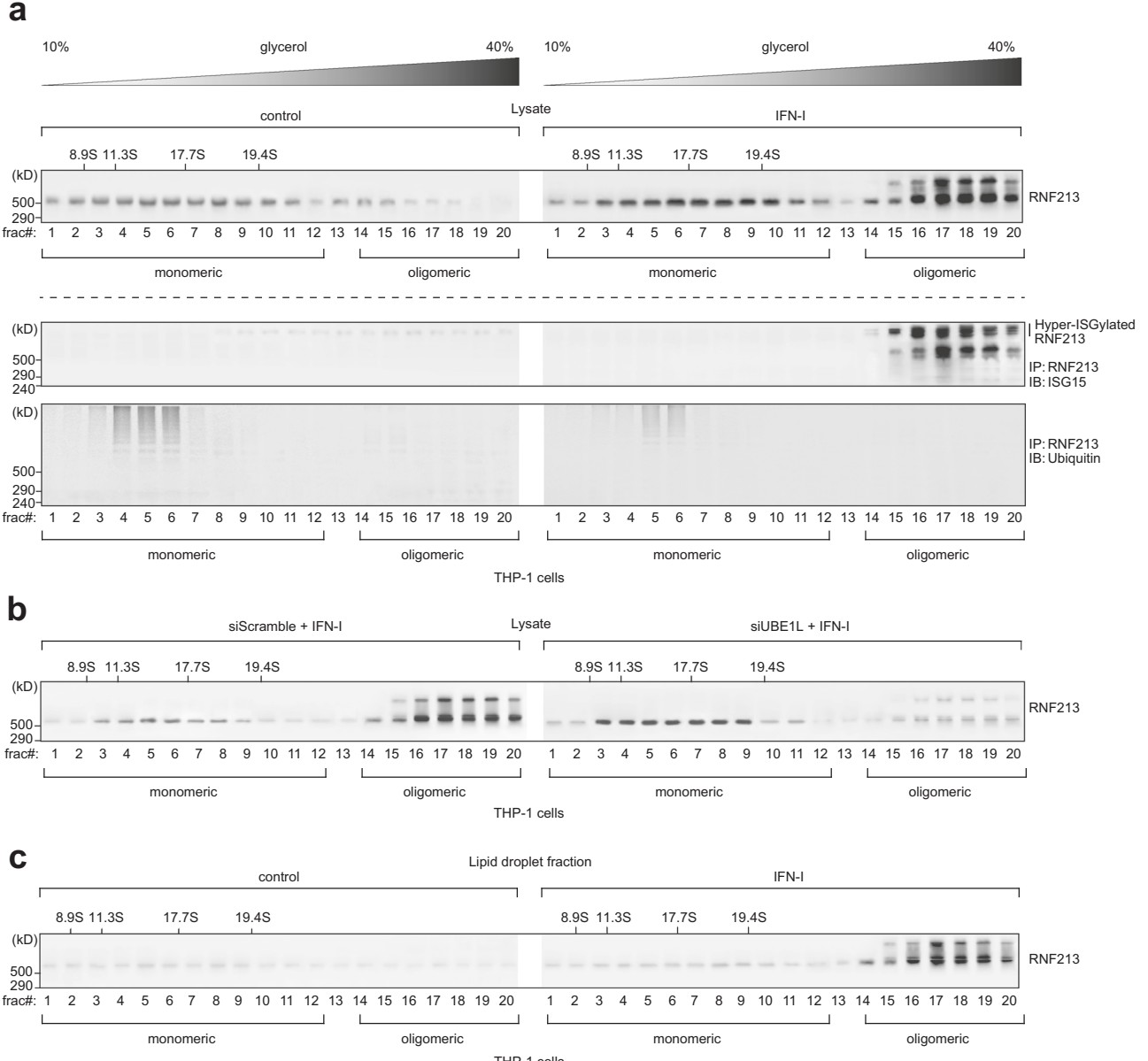

**Fig. 3 Type I interferon induces RNF213 ISGylation and oligomerization on lipid droplets. a** THP-1 cells were treated with interferon (IFN)-β for 8 h or left untreated. Lysates were separated by density gradient ultracentrifugation on glycerol gradients (10–40% (v/v), Svedberg constants of the standard markers are indicated above the blots) to isolate the monomeric versus oligomeric form of RNF213. Twenty fractions (frac#) for each sample were collected, concentrated by TCA precipitation and analyzed by immunoblotting against RNF213, showing the presence of oligomer RNF213 in fraction 14–20 upon interferon treatment (upper panel). Alternatively, RNF213 was immunoprecipitated (IP) from each fraction, first desalted over Amicon columns. Immunoprecipitated material was eluted into loading buffer and analyzed by immunoblotting against ISG15 and ubiquitin, showing (hyper)ISGylation of oligomer RNF213 upon interferon treatment (lower panel). **b** The monomeric and oligomeric forms of RNF213 were separated by density gradient ultracentrifugation after interferon-β treatment as in **a** and knockdown of UBE1L by siRNA (siUBE1L) treatment for 48 h, using a nontargeting scrambled siRNA (siScramble) as control. Knockdown of UBE1L strongly reduced RNF213 oligomerization upon interferon treatment. **c** LDs were isolated by ultracentrifugation floatation assay on a sucrose step-gradient and associated proteins were further separated by density gradient ultracentrifugation to isolate the monomeric and oligomeric form of RNF213 after interferon-β treatment as in **a**. Fractions were concentrated by TCA precipitation and analyzed by immunoblotting against RNF213, showing association of oligomeric RNF213 with LDs upon interferon treatment. Source data are provided as a Source Data file.

fraction followed by immunoblotting for ISG15 showed that this was indeed the case, while a ubiquitin-specific signal was only detected in the low-density fractions (Fig. 3a). The ISG15 blot revealed two major groups of ISGylated RNF213 with one group of bands running at virtually the same height as monomeric RNF213 and a second, hyper-ISGylated group of bands running well above (Fig. 3a). Knockdown of UBE1L, the E1 activating

enzyme for ISG15, strongly reduced the intensity of both groups of ISGylated RNF213 while it increased the intensity of monomeric RNF213, indicating that ISGylation of RNF213 is required for its oligomerization (Fig. 3b and Supplementary Fig. 6c). Finally, we repeated the glycerol sedimentation experiment after LD isolation and showed that the oligomeric, ISGylated forms of RNF213 are associated with LDs only after IFN-I treatment

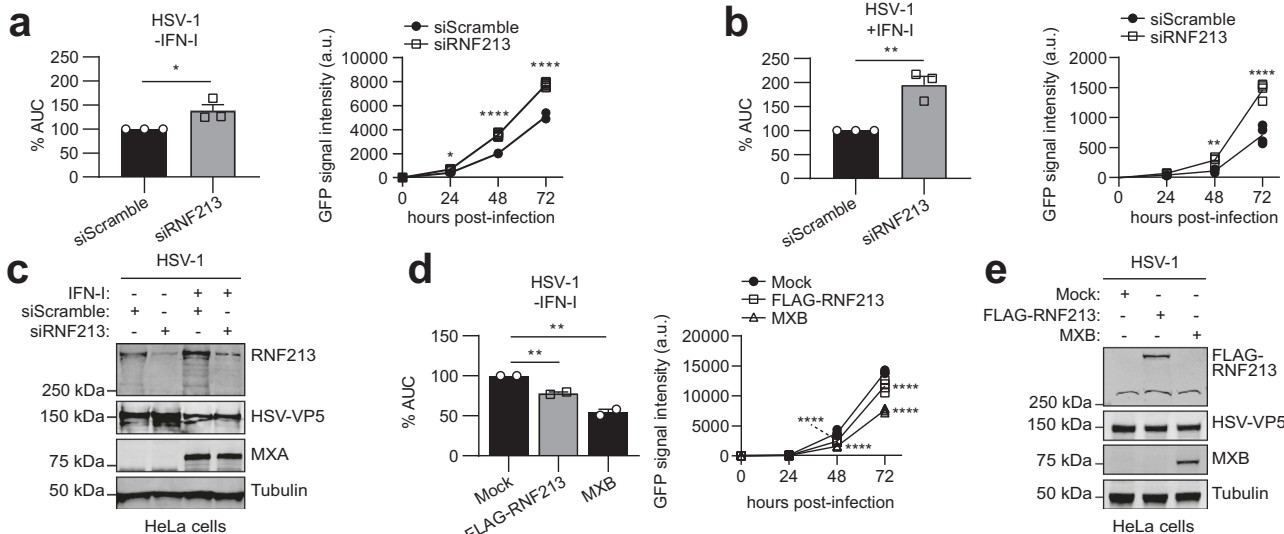

**Fig. 4 RNF213 counteracts HSV-1 infection. a** HeLa cells were infected up to 72 h with eGFP-expressing recombinant herpes simplex virus 1 (HSV-1) at MOI 0.1. Forty eight hours prior to infection, cells were transfected with a pool of siRNAs targeting RNF213 (siRNF213) or a pool of scrambled siRNAs (siScramble) as control. The viral load was determined by monitoring the GFP signal in each condition every 24 h to generate a viral growth curve (right panel, representative viral growth curve from a single experiment, $n = 4$ technical replicates, curve connecting AVG, two-tailed unpaired $t$-test comparing siRNF213 to siScramble, a.u. arbitrary units). The area under the curve (AUC) was calculated for each growth curve and the average AUC of three independent experiments is shown relative to the siScramble control (left panel, AVG ± SEM, $n = 3$ independent experiments, two-tailed unpaired $t$-test). **b** HSV-1 infection experiment performed as in **a**, except that 16 h prior to infection cells were treated with interferon (IFN)-α (right panel, representative viral growth curve from a single experiment, $n = 4$ technical replicates, curve connecting AVG, two-tailed unpaired $t$-test comparing siRNF213 to siScramble; left panel, AVG ± SEM, $n = 3$ independent experiments, two-tailed unpaired $t$-test). Knockdown of RNF213 leads to significantly higher HSV-1 infection levels. **c** Immunoblots against RNF213, HSV-VP5, and MXA with tubulin as loading control confirmed knockdown of RNF213, HSV-1 infection and interferon-α treatment, respectively, in the experiments shown in **a, b. d** HSV-1 infection experiment performed as in **a**, except that 24 h prior to infection at MOI 0.05 cells were transfected with plasmids encoding 3xFLAG-RNF213 or MXB or with an empty vector (mock) as control (right panel, representative viral growth curve from a single experiment, AVG ± SEM, $n = 4$ technical replicates, curve connecting AVG, two-tailed unpaired $t$-test comparing RNF213 or MXB overexpression to mock). The average AUC of two independent experiments is shown relative to the mock control (left panel, AVG ± SEM, $n = 2$ independent experiments, two-tailed unpaired $t$-test). Overexpression of RNF213 leads to significantly lower HSV-1 infection levels. **e** Immunoblots against FLAG, HSV-VP5, and MXB with tubulin as loading control confirmed HSV-1 infection and expression of FLAG-RNF213 and MXB in the experiments shown in **d**. In **a, b, e** asterisks indicate $p$ values with *$p < 0.05$, **$p < 0.01$, and ****$p < 0.0001$. Source data are provided as a Source Data file.

(Fig. 3c). Together, these results show that IFN-I induces the oligomerization of RNF213 on LDs and that this process requires ISG15 modification of RNF213 itself.

**RNF213 exerts broad antimicrobial activity in vitro.** The discovery of RNF213 as a sensor for ISGylated proteins made us suspect that RNF213 could also have antiviral properties, similar to ISG15, which was in line with the recently reported increased susceptibility of *Rnf213*-deficient mice to Rift Valley fever virus infection[62]. To investigate this, we infected cells with reduced or enhanced expression levels of RNF213 with different viruses. RNF213 knockdown or overexpression did not affect ISG expression (Supplementary Fig. 7a, b). We first infected HeLa cells with herpes simplex virus 1 (HSV-1), a widespread human pathogen that can cause cold sores and genital herpes that is ISG15 sensitive[63]. We used a GFP-expressing HSV-1 strain to monitor infection levels up to 3 days post infection by fluorescence intensity. Knockdown of RNF213 led to a small but significant increase in infection compared to scrambled siRNA control, both in the presence or absence of IFN-I (Fig. 4a–c). Conversely, overexpression of FLAG-RNF213 significantly lowered HSV-1 infection levels, although not to the same extent as overexpressed MXB, which was recently reported as a Herpesvirus restriction factor (Fig. 4d, e)[64]. To test whether the antiviral effect of RNF213 depends on ISG15, we repeated the RNF213 knockdown experiment in ISG15 KO HeLa cells. Similar as in WT cells depletion of RNF213 led to higher HSV-1 infection

levels, indicating that restriction of HSV-1 by RNF213 does not require ISG15 (Supplementary Fig. 7c-h). We also determined the possible involvement of RNF213 in the control of replication of human respiratory syncytial virus (RSV), an important human respiratory pathogen that is susceptible to an ISG15-dependent antiviral effect[10]. For these experiments, A549 cells were infected with RSV-A2 at a low multiplicity of infection (MOI) and incubated for 6 days to allow multiple rounds of infection (Supplementary Fig. 8a). Knockdown of RNF213 led to a small but significant increase in the viral load in the cell culture supernatants 5 days post infection compared to scrambled siRNA control, an effect that disappeared at day 6 (Fig. 5a). In addition, when the cells were treated with IFN-I, significantly higher viral loads were observed in siRNF213-treated cells compared to siScramble-treated cells at day 5 and 6 after infection (Fig. 5b). We also assessed the effect of RNF213 knockdown on RSV replication in a plaque assay because the plaque size is a measure for the extent of cell-to-cell spread of the virus. On day 6 after RSV infection, we observed significantly larger plaques upon knockdown of RNF213 both with and without IFN-I, suggesting also increased cell-to-cell spread of RSV under these conditions (Fig. 5c, d). As a third ISG15-sensitive virus we tested CVB3, a Picornavirus that can lead to cardiomyopathy and that is known to be ISG15 sensitive[11,12] (Supplementary Fig. 8b). Again, knockdown of RNF213 significantly increased the replication of CVB3 as indicated by higher viral genome (Fig. 6a) and enhanced viral protein expression levels (Fig. 6b) as well as by the elevated

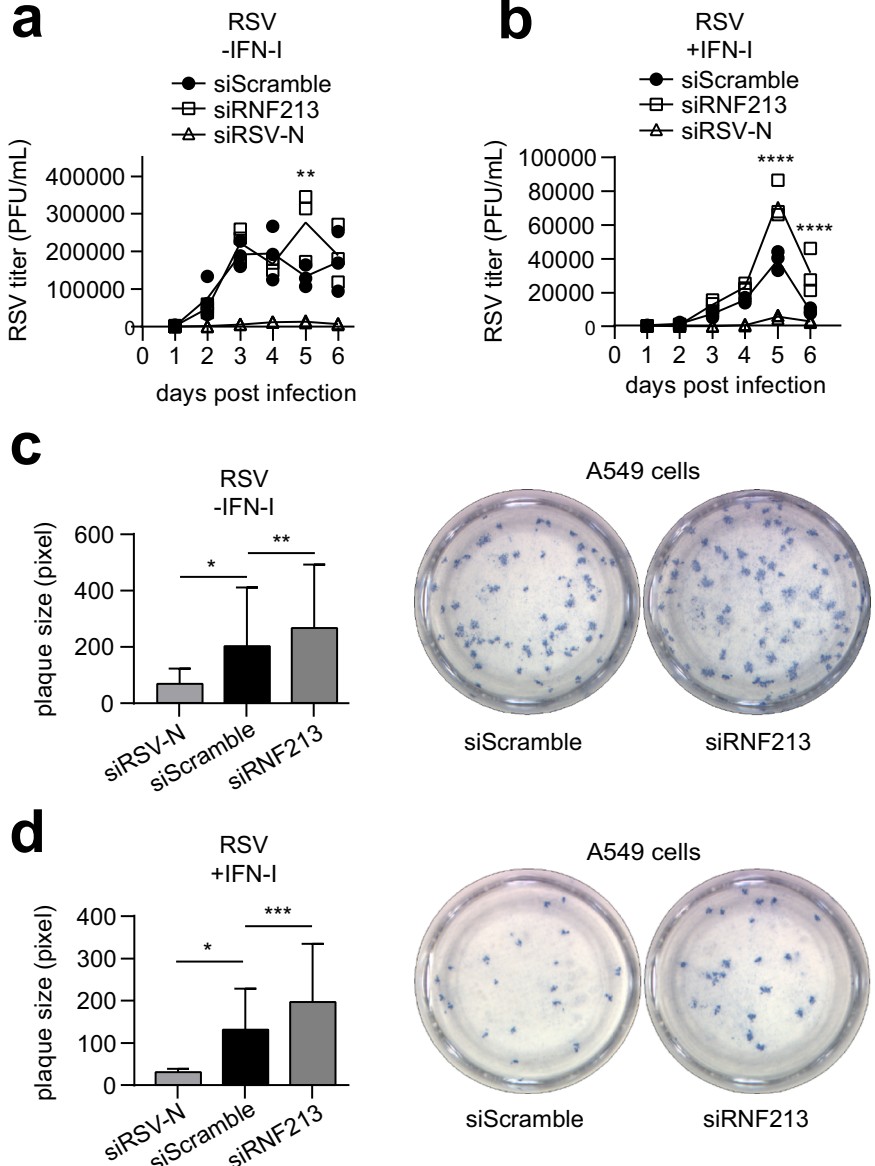

**Fig. 5 RNF213 counteracts RSV infection. a** A549 cells were infected with human respiratory syncytial virus (RSV) A2 for up to 6 days at MOI 0.02. Forty eight hours prior to infection, cells were transfected with a pool of siRNAs targeting RNF213 (siRNF213), a single siRNA targeting the RSV-nucleoprotein (siRSV-N) as positive control or a pool of scrambled siRNAs (siScramble) as negative control. The viral titer was determined by counting plaque-forming units (PFUs) after serial dilution (representative results from a single experiment, AVG ± SEM, $n = 3$ technical replicates, two-tailed unpaired $t$-test comparing siRNF213 to siScramble). **b** RSV infection experiment performed as in **a**, except that 42 h prior to infection cells were treated with interferon (IFN)-β (representative results from a single experiment, AVG ± SEM, $n = 3$ technical replicates, two-tailed unpaired $t$-test comparing siRNF213 to siScramble). Knockdown of RNF213 leads to significantly higher RSV titers. **c** A549 cells were infected with RSV-A2 at MOI 0.005 in combination with knockdown of RNF213 and RSV-N as described in **a**. Six days post infection, a plaque assay was performed and plaque sizes were quantified in pixels with Fiji (left panel, representative results from a single experiment, AVG ± SD, two-tailed Mann–Whitney test, siRSV-N $n = 8$, siScramble $n = 267$, and siRNF213 $n = 183$). Representative images showing plaques of siRNF213 and siScramble-treated cells (right panel). **d** RSV infection experiment performed as in **c**, except that 42 h prior to infection cells were treated with interferon-β (left panel, representative results from a single experiment, AVG ± SD, two-tailed Mann–Whitney test, siRSV-N $n = 4$, siScramble $n = 123$, and siRNF213 $n = 99$). Representative images showing plaques of siRNF213 and siScramble-treated cells (right panel). Knockdown of RNF213 leads to significantly larger RSV plaques. In **a–d**) asterisks indicate $p$ values with *$p < 0.05$, **$p < 0.01$, ***$p < 0.001$, and ****$p < 0.0001$. Source data are provided as a Source Data file.

formation of infectious viral particles from cells (Fig. 6c). Conversely, overexpression of RNF213 resulted in reduced infection (Supplementary Fig. 8c-e).

We subsequently wondered whether RNF213 could be broadly antimicrobial and also target bacterial pathogens. Thus, we infected HeLa cells with *Listeria monocytogenes* (further referred to as *Listeria*), a facultative intracellular bacterium that was

recently reported to be counteracted by ISG15[6] (Supplementary Fig. 9a). When we knocked down RNF213 with a pool of siRNAs, we measured a 30% increase in intracellular bacteria compared to a pool of control siRNAs, an increase that was similar to what we observed by knockdown of ISG15 (Fig. 7a). Individual siRNAs against RNF213 also augmented the bacterial load (Supplementary Fig. 9b) and we observed no effect of RNF213 knockdown or

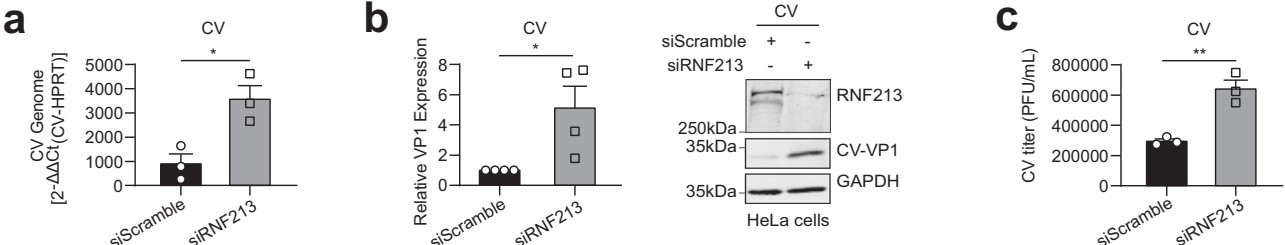

**Fig. 6 RNF213 counteracts CVB3 infection. a** HeLa cells were infected with coxsackievirus (CV) B3 at MOI 0.01. Twenty four hours prior to infection, cells were transfected with a pool of siRNAs targeting RNF213 (siRNF213) or a pool of scrambled siRNAs (siScramble) as control. Twenty four hours post infection, the intracellular viral RNA load was determined by qRT-PCR (AVG ± SEM, $n = 3$ independent experiments, two-tailed unpaired $t$-test). **b** From the same experiment as in **a**, the intracellular viral protein load was determined by immunoblotting against VP1 and the intensity of the VP1 band is shown relative to the siScramble control (left panel, AVG ± SEM, $n = 4$ independent experiments, one-tailed $t$-test). A representative immunoblot for the quantification of VP1 is shown (right panel). **c** From the same experiment as in **a**, the viral titer was determined by counting PFUs after serial dilution (representative result from a single experiment, AVG ± SEM, $n = 3$ technical replicates, two-tailed unpaired $t$-test). Knockdown of RNF213 leads to a significant increase in CVB3 infection. In **a–c** asterisks indicate $p$ values with *$p < 0.05$, **$p < 0.01$. Source data are provided as a Source Data file.

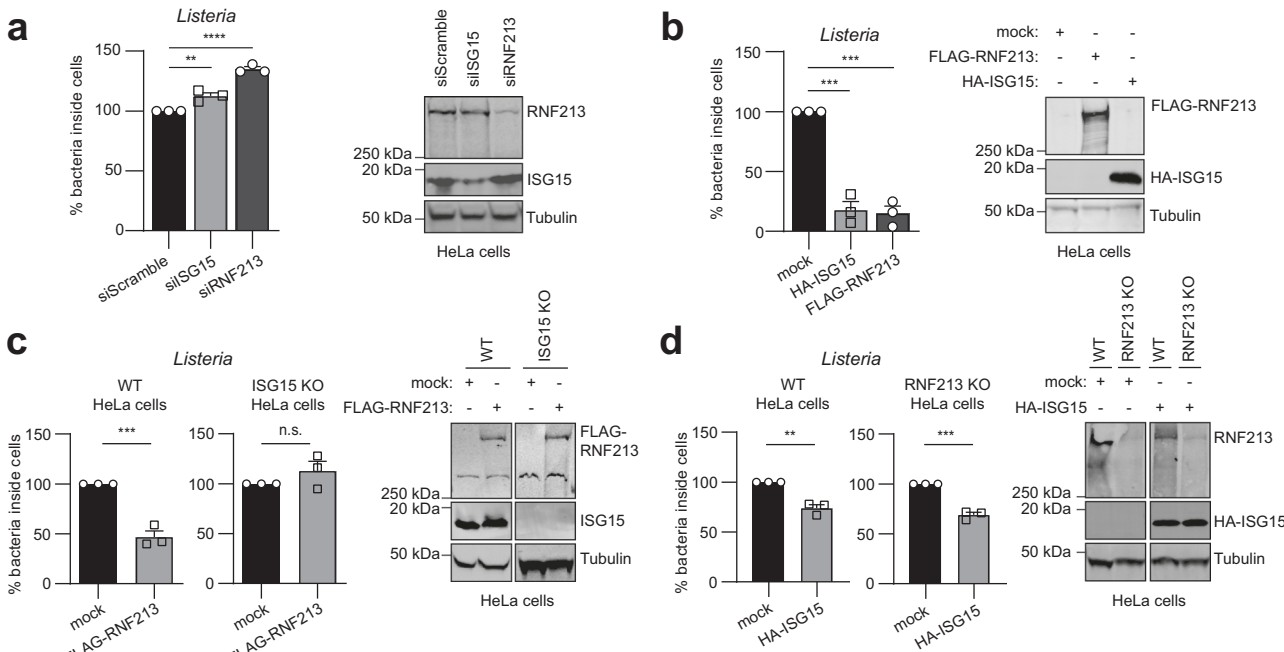

**Fig. 7 RNF213 counteracts *Listeria* infection. a** HeLa cells were infected with *Listeria monocytogenes* EGD (*Listeria*) for 16 h at MOI 25. Twenty four hours prior to infection, cells were transfected with a pool of siRNAs targeting ISG15 (siISG15), RNF213 (siRNF213), or a pool of scrambled siRNAs (siScramble) as control. Intracellular *Listeria* were quantified after serial dilution by counting colony-forming units (CFUs) in a gentamycin assay. The percentage of intracellular bacteria relative to siScramble-transfected cells is shown (left panel, AVG ± SEM, $n = 3$ independent experiments, two-tailed unpaired $t$-test). Immunoblots against RNF213 and ISG15 with tubulin as loading control (right panel). **b** HeLa cells were infected with *Listeria* for 4 h at MOI 25. Twenty four hours prior to infection, HeLa cells were transfected with plasmids encoding 3xFLAG-RNF213 or HA-ISG15 or with an empty vector (mock) as control. Intracellular *Listeria* were quantified as in **a** and the percentage of intracellular bacteria relative to mock plasmid-transfected cells is shown (left panel, AVG ± SEM, $n = 3$ independent experiments, two-tailed unpaired $t$-test). Immunoblots against FLAG and HA with tubulin as loading control (right panel). **c** Wild-type (WT) or ISG15 knockout (KO) HeLa cells were infected with *Listeria* for 16 h at MOI 25 after transfection of FLAG-RNF213 as in **b**. Intracellular *Listeria* were quantified as in **a** and the percentage of intracellular bacteria relative to mock plasmid-transfected cells is shown (left panel, AVG ± SEM, $n = 3$ independent experiments, two-tailed unpaired $t$-test, n.s. not significant). Immunoblots against FLAG and ISG15 with tubulin as loading control (right panel). **d** Wild-type (WT) or RNF213 KO HeLa cells were infected with *Listeria* for 16 h at MOI 25 after transfection of HA-ISG15 as in **b**. Intracellular *Listeria* were quantified as in **a** and the percentage of intracellular bacteria relative to mock plasmid-transfected cells is shown (left panel, AVG ± SEM, $n = 3$ independent experiments, two-tailed unpaired $t$-test). Immunoblots against RNF213 and HA with tubulin as loading control (right panel). In **a–d** asterisks indicate $p$ values with **$p < 0.01$, ***$p < 0.001$, and ****$p < 0.0001$. Source data are provided as a Source Data file.

overexpression on bacterial entry (Supplementary Fig. 9c, d). Conversely, overexpression of FLAG-RNF213 reduced the bacterial load with 85% relative to mock transfected cells, a reduction that was similar to overexpression of HA-ISG15 (Fig. 7b). Importantly, the protective effect of overexpressed FLAG-RNF213 was lost in ISG15 knockout cells (Fig. 7c),

indicating that, unlike for the antiviral activity, the antimicrobial activity of RNF213 requires ISG15. In contrast, overexpressed HA-ISG15 still counteracted *Listeria* in RNF213 knockout (Fig. 7d) or knockdown cells (Supplementary Fig. 9e), meaning that ISG15 also harbors antimicrobial activity independent of RNF213. Taken together, we showed that increasing the

expression levels of RNF213 led to lower infection levels of HSV-1, CVB3, and *Listeria* in cultured cells, while reducing the levels of RNF213 promoted in vitro infection with these pathogens as well as RSV. Compared to the antiviral activity, the antibacterial effect of RNF213 was more pronounced and dependent on ISG15. These results show that RNF213 plays an important role in the innate cellular immune response against invading *Listeria*, which is in line with recent findings on *Salmonella*[65] and indicating a functional link between ISG15 and RNF213.

**RNF213 decorates intracellular *Listeria* and is antibacterial in vivo**. Given the protective function of RNF213 against *Listeria*, we wondered if RNF213 adopts a specific subcellular localization during infection. To this end, we transfected HeLa cells with eGFP-RNF213. Co-staining for LDs followed by fluorescence microscopy revealed that RNF213 colocalized with the majority (~70%) of LDs, both in uninfected and *Listeria*-infected cells (Fig. 8a, b). We noticed that infection with *Listeria* was associated with an increase in LDs (Supplementary Fig. 10a), similar to previous results in human macrophages, which had phagocytosed *E. coli* or *Salmonella*[66]. Intriguingly, RNF213 also colocalized with a subset of intracellular *Listeria*, and seemed to decorate the bacterial surface (Fig. 8a). To quantify how many bacteria were targeted, we repeated the experiment with mCherry-expressing *Listeria* and found that on average approximately 40% of intracellular *Listeria* cocolocalized with RNF213 at 18 h post infection (Fig. 8c, d). In order to determine whether RNF213 was docking on bacteria or on membrane remnants near bacteria, we made use of a triple mutant, which lacks phospholipase A (PlcA), phospholipase B (PlcB), and the primary hemolysin Listeriolysin O (LLO)[6]. This mutant is unable to escape from the vacuole of epithelial cells and when compared with the parental strain at 6 h post infection, we did not observe colocalization with RNF213 (Fig. 8e), indicating that RNF213 likely targets a subset of cytosolic bacteria, as was recently shown for *Salmonella*[65]. The latter study demonstrated that ubiquitination of cytosolic *Salmonella* by RNF213 initiates autophagic clearance. Unlike *Salmonella*, however, the majority of wild-type cytosolic *Listeria* evades autophagy through a variety of mechanisms including the use of ActA to sterically protect the surface of bacteria from being modified by ubiquitin[67] and of PlcA to alter phospholipid levels in the cell to thereby interfere with LC3-associated phagocytosis in macrophages or autophagy initiation in epithelial cells[68,69]. Furthermore, more recent work suggests that *Listeria* can evade host autophagy in a multistep process involving LLO, ActA, and PlcA[70]. As such, we visualized actin using phalloidin and mono or poly ubiquitin linkages using anti-FK2 in RNF213 over-expressing, *Listeria*-infected cells. Our data are in line with the published role of ActA as surfaces of individual bacteria marked by RNF213 and ubiquitin (sides of bacteria and sites of division) did not colocalize with surfaces marked by actin once ActA has relocalized to the bacterial poles (Fig. 8e). Furthermore, RNF213 and ubiquitin colocalize on the bacterial surface suggesting that similar to its modification of *Salmonella*, RNF213 also acts as a ubiquitin E3 ligase during *Listeria* infection (Supplementary Fig. 10b). Together, these data suggest that overexpression of RNF213 prior to infection can overcome the capacity of wild-type *Listeria* to evade the initial host-mediated targeting steps that mark the bacterial surface with ubiquitin and allow the host cell to ultimately dispose of bacteria in an autophagosome or auto-lysosome. Future work will assess the interplay of RNF213 and downstream autophagic targeting of *Listeria* in both phagocytic and epithelial cells.

In order to test whether this localization has functional consequences in vivo, we deleted RNF213 in mice using CRISPR/Cas9 targeting of mouse *RNF213* exon 28 as previously described[59] (Supplementary Fig. 11). At 24, 48, and 72 h post infection with *Listeria* we observed a dramatic increase in bacterial burden in the spleen of $Rnf213^{-/-}$ animals compared to WT littermate controls, which increased over time (Fig. 9a). At 72 h, a significant increase was also detected in the liver (Fig. 9b) and this observation was confirmed in two additional independent experiments (Supplementary Fig. 12a-d). This profound difference in bacterial load, particularly in the spleen, reveals a central role of RNF213 in the host defense against bacterial infection. Compared to WT littermates, RNF213 KO animals were also more likely to lose more bodyweight and reach the ethical endpoint following challenge with a mouse-adapted strain of RSV even though the difference in lung virus loads determined on day 5 after infection did not reach statistical significance (Supplementary Fig. 12e-g). Future work will assess which cell type is unable to control *Listeria* infection and whether RNF213 deletion also affects adaptive immune responses in addition to innate responses. This is of particular relevance to understanding the role of inflammation or prior infection in human patients with inborn errors in RNF213.

**The RNF213 E3 module is required for its antibacterial activity**. Ahel et al. recently published the cryo-EM structure of monomeric RNF213[71] showing that RNF213 does not depend on its RING domain to function as an E3 ubiquitin ligase, but instead classifies as a special type of E3 ligase employing a Cys-containing motif to promote ubiquitin transfer via transthiolation. While a ΔRING still exhibits ubiquitination activity[65,71] and without knowledge of the catalytic residues, we sought to use this structural data to test whether the E3 ligase activity of RNF213 is required for its antimicrobial function. We therefore deleted the C-terminal 1210 amino acids of RNF213 comprising the large E3 module and small C-terminal domain to generate a FLAG-RNF213ΔC construct that is expected to be devoid of any ubiquitination activity. GST pull-down showed that RNF213ΔC was still capable of binding ISG15, while the complementary C-terminal fragment was not (Fig. 10a). Similarly, FLAG-RNF213ΔC was capable of pulling down a smear of ISGylated proteins to the same extent as full-length FLAG-RNF213 (Fig. 10b). However, without any effects on bacterial entry (Supplementary Fig. 9c), overexpression of RNF213ΔC no longer retained its notable antimicrobial effects when overexpressed as it could not reduce the amount of intracellular *Listeria* in infected HeLa cells (Fig. 10c). Furthermore, fluorescence microscopy confirmed that eGFP-RNF213ΔC no longer colocalizes with intracellular *Listeria* (Fig. 10d) or with LDs (Supplementary Fig. 13a-b). Taken together, these results indicate that the E3 module of RNF213, and likely its ubiquitination activity, is required for its bacterial targeting and antimicrobial activity. Of note, while the present paper was in review, Otten et al. reported a catalytic zinc-binding RZ domain adjacent to RNF213's RING domain[65], something that is mechanistically supported by recent structural work[72]. In line with the data on *Listeria* presented here, catalytic activity of the RZ domain was shown to be required for targeting of RNF213 to cytosolic *Salmonella*[65].

## Discussion
We report the discovery of RNF213 as an intracellular sensor for proteins modified by ISG15, a ubiquitin-like, immunity-related modification. We show that IFN-I signaling induces the oligomerization and translocation of RNF213 on lipid droplets in macrophages and that this process requires ISGylation of RNF213 itself. Furthermore, in vitro and in vivo infection assays with four different pathogens revealed an as yet undescribed, broad

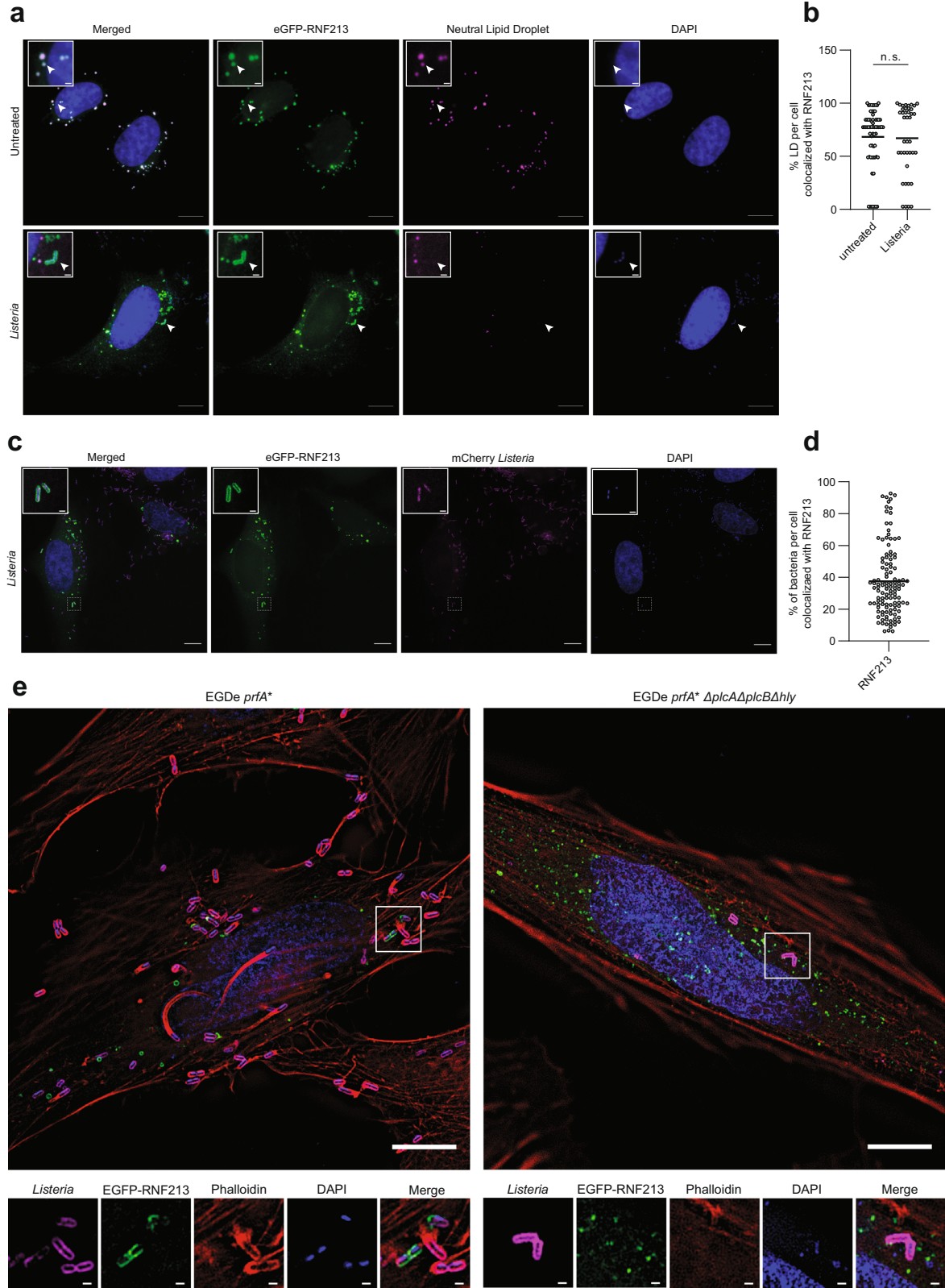

antimicrobial function of RNF213. Thus, our unique Virotrap method revealed a functional link between ISG15 and RNF213, as well as an undiscovered role for RNF213 in host defense pathways to both bacterial and viral infections.

We identified RNF213 as an ISG15-binding protein by Virotrap and confirmed the interaction by GST pull-down assays, methods that both create a curved surface, either convex or concave, studded with ISG15 molecules which could be a surrogate for multiply modified proteins or larger binding surfaces such as organelles. It was of particular interest, that we identified a large AAA + ATPase as ISG15 interactor under these conditions, RNF213, which presumably docks on ISG15 surfaces or vice versa. Indeed, our data suggest that oligomeric RNF213 acts as a binding platform for ISGylated proteins on the surface of

**Fig. 8 RNF213 decorates intracellular *Listeria*. a** Representative images of HeLa cells transfected with eGFP-RNF213 and counterstained for lipid droplets (LDs). Following transfection cells were left untreated or infected with *Listeria monocytogenes* EGD for 24 h at MOI 25. Scale bars in the pictures and insets are respectively 10 microns and 1 micron. White arrows indicate colocalization between RNF213 and lipid droplets or instances of RNF213 colocalization with intracellular bacteria (DAPI = 4′,6-diamidino-2-phenylindole). **b** LDs in uninfected (*n* = 66 cells) and *Listeria*-infected cells (*n* = 40 cells) from **a** were quantified with Fiji and the percentage of LDs per cell that colocalized with RNF213 was calculated (representative results from a single experiment, two-tailed unpaired *t*-test, AVG uninfected = 68.29%, AVG Listeria-infected = 67.14%, n.s. = not significant). **c** Representative images of HeLa cells transfected with eGFP-RNF213 and infected for 18 h with *Listeria monocytogenes* EGD stably expressing mCherry. Scale bars in the pictures and insets are respectively 10 microns and 0.5 micron. **d** Intracellular *Listeria* from **c** were quantified with Imaris 9.6 and the percentage of *Listeria* that was decorated by eGFP-RNF213 was calculated for each field by mapping the cell surface, enumerating intracellular bacteria and quantifying bacteria that colocalized with GFP-RNF213 (defined as bacteria within 0.5 μm of RNF213, see methods for more detail). At least 200 cells were counted per experiment, and data were compiled from three independent experiments indicating that on average 37.44% of *Listeria* was decorated by RNF213. **e** (left panel) Representative image of HeLa cells transfected with eGFP-RNF213 and infected with *Listeria monocytogenes* EGDe *prfA*\* at MOI 1 for 6 h. Scale bar in the picture and insets is 10 microns. Inset demonstrates colocalization of RNF213 to the surface of cytosolic bacteria. Nuclei are shown in blue, RNF213 in green, actin in red, and bacteria in magenta. Scale bar is 1 micron. (right panel) Representative image of HeLa cells transfected with eGFP-RNF213 and infected with *Listeria monocytogenes* EGDe *prfA*\**ΔplcA*Δ*plcB*Δ*hly* at MOI 75 for 6 h. Scale bars in the picture and insets are respectively 10 microns and 1 micron. Source data are provided as a Source Data file.

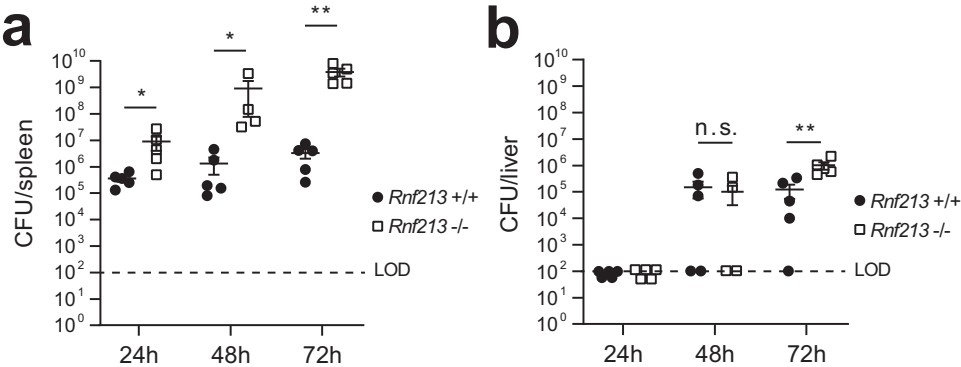

**Fig. 9 RNF213 counteracts *Listeria* infection in vivo. a, b** *Rnf213*−/− and *Rnf213*+/+ mice were infected intravenously with 5 × 10⁵ *Listeria monocytogenes* EGD. Spleen (**a**) and liver (**b**) were isolated following 24 h (*n* = 5 for both genotypes), 48 h (*n* = 4 for *Rnf213*−/− and *n* = 5 for *Rnf213*+/+), and 72 h of infection (*n* = 5 for both genotypes). Colony-forming Units (CFUs) per organ were counted by serial dilution and replating; dots and squares depict individual animals (representative results of a single experiment, AVG ± SEM, two-tailed Mann–Whitney test, n.s. not significant, LOD limit of detection). *Rnf213*−/− mice are dramatically more susceptible to *Listeria* as evidenced by significantly higher CFUs at all three timepoints in the spleen and at 72 h in the liver. The results of two independent repeat experiments are shown in Supplementary Fig. 12. Asterisks indicate *p* values with \**p* < 0.05 and \*\**p* < 0.01. Source data are provided as a Source Data file.

lipid droplets. In line with this, we demonstrated that ISGylated proteins co-immunoprecipitated with RNF213 and that fewer ISGylated proteins were isolated on lipid droplets without RNF213. It remains unknown what happens to ISGylated proteins after binding to RNF213. As a model, we were inspired by similarities between the interaction of poly-SUMOylated substrate proteins with RNF4, a SUMO-targeted ubiquitin ligase (STUbL)[51] that could be an analogous system to RNF213 and ISGylation. In the STUbL pathway, nuclear stresses lead to poly-SUMOylation of protein targets that are further recognized by RNF4. RNF4 then ubiquitinates the poly-SUMOylated proteins, targeting them for degradation by the proteasome. Our model is that RNF213 might recognize multi-ISGylated proteins as the entry point of a pathway for further processing of multi-ISGylated proteins at lipid droplets. While RNF4 and poly-SUMOylation are both induced by nuclear stress, RNF213 and multi-ISGylation are induced by IFN-I signaling[60,73]. Moreover, co-regulation of ISG15 and RNF213 was previously observed in a large-scale analysis of protein quantitative trait loci (pQTL)[74], also suggesting that both proteins operate in the same cellular pathway. Our future work will address the fate of ISGylated proteins that bind RNF213. It is particularly intriguing to determine if the E3 module of RNF213 ubiquitinates the bound ISGylated proteins for

proteasomal degradation, which could be substantiated by the identification of VCP and proteasome components alongside RNF213 in the ISG15 Virotrap experiments. Alternatively, the ISGylated proteins bound to RNF213 might be degraded via lipophagy[75] or targeted for an alternative fate.

RNF213 is a very large, poorly characterized human protein. Recent functional studies highlighted the role of RNF213 in lipid metabolism, mediating lipid droplet formation[57], and lipotoxicity[76]. The latter study also assessed changes in the RNF213-dependent ubiquitylome and showed that RNF213 was required to activate the nuclear factor κB (NFκB) pathway upon palmitate treatment. This NFκB-inducing function of RNF213 would, however, be negatively regulated by its ubiquitin ligase activity[77] and requires further investigation, especially in light of the RNF213 antimicrobial activity reported here. The recently published cryo-EM structure of monomeric RNF213 revealed three major structural components comprising an N-terminal stalk, a dynein-like ATPase core and a C-terminal E3 module[71]. Previous studies showed that the ATPase activity of RNF213 is essential for its multimerization and localization to lipid droplets; however, in contrast to most AAA + ATPases that form stable hexamers[55], multimerization of RNF213 is proposed to be a dynamic process, similar to the dynamic

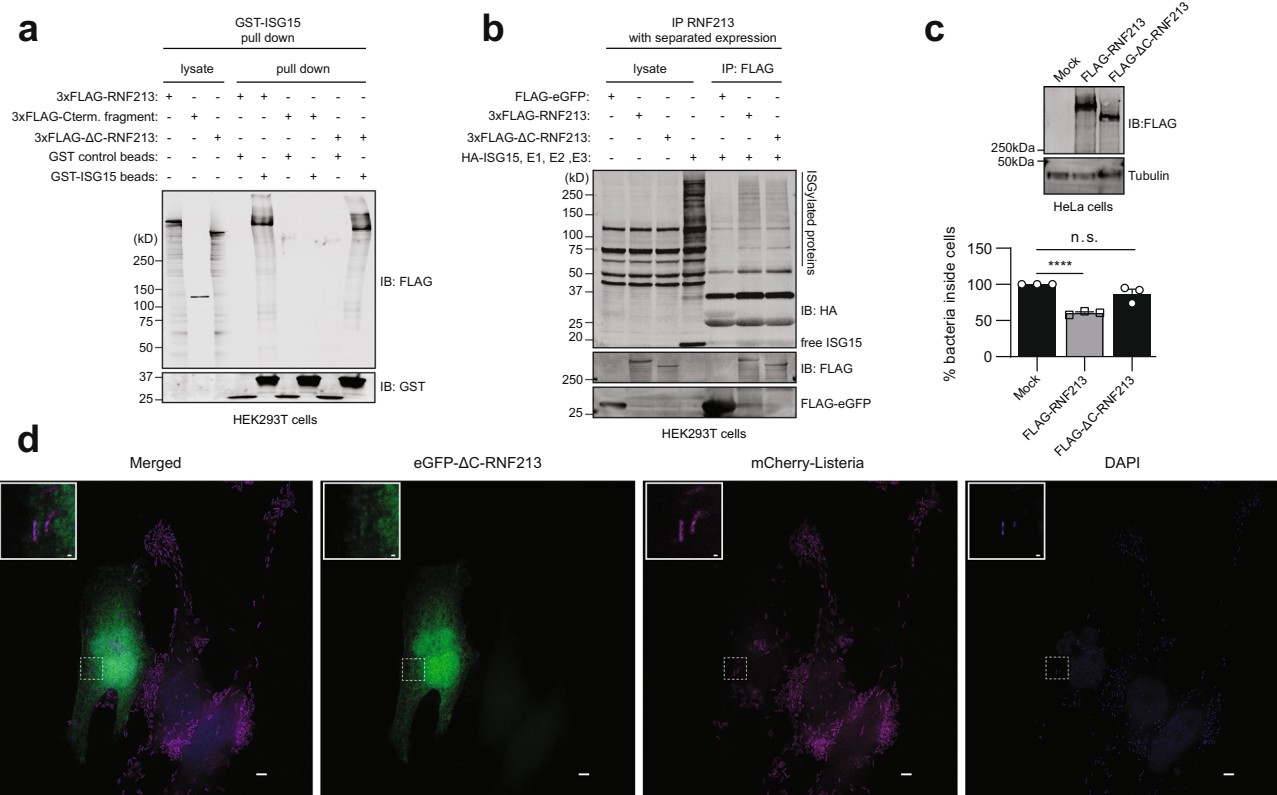

**Fig. 10 The RNF213 E3 module is required for its antimicrobial activity. a** Glutathione-S-Transferase (GST) pull-down with ISG15 followed by immunoblotting shows that 3xFLAG-RNF213ΔC binds to ISG15 similar to full-length 3xFLAG-RNF213. Beads coated with GST-ISG15 were mixed with a lysate of HEK293T cells expressing full-length 3xFLAG-RNF213, 3xFLAG-RNF213ΔC, or the complementary C-terminal fragment of RNF213 and bound proteins were analyzed by immunoblot (IB) against FLAG and GST. **b** FLAG immunoprecipitation (IP) was performed from lysates of HEK293T cells expressing 3xFLAG-RNF213, 3xFLAG-RNF213ΔC, or FLAG-eGFP as control. After immunoprecipitation, beads were mixed with a lysate of HEK293T cells expressing HA-ISG15 and the ISGylation machinery (E1, E2, E3). 3xFLAG-RNF213ΔC was capable of pulling down ISGylated proteins similar to 3xFLAG-RNF213, while FLAG-eGFP was not. **c** HeLa cells were infected with *Listeria monocytogenes* EGD for 4 h at MOI 25. Twenty four hours prior to infection, HeLa cells were transfected with plasmids encoding 3xFLAG-RNF213 or 3xFLAG-RNF213ΔC or with an empty vector (mock) as control. Intracellular *Listeria* were quantified after serial dilution by counting colony-forming units (CFUs) in a gentamycin assay. The percentage of intracellular bacteria relative to mock plasmid-transfected cells is shown (lower panel, AVG ± SEM, $n = 3$ independent experiments, two-tailed unpaired *t*-test, n.s. not significant). Immunoblots against FLAG with tubulin as loading control confirmed expression of FLAG-RNF213, FLAG-RNF213ΔC (upper panel). **d** Representative images of HeLa cells transfected with eGFP-RNF213ΔC and infected for 18 h with *Listeria monocytogenes* EGD stably expressing mCherry. Scale bars in the pictures and insets are respectively 10 microns and 0.5 micron. eGFP-RNF213ΔC showed a diffused cellular staining, not decorating intracellular *Listeria* (DAPI = 4′,6-diamidino-2-phenylindole). In **c** asterisks indicate *p* values with ****$p < 0.0001$. Source data are provided as a Source Data file.

assembly of the microtubule-severing protein katanin[55,57]. In this model RNF213 exists as a monomer in solution and forms a hexamer on the surface of lipid droplets upon ATP binding[57]. Contrary to this, the RNF213 structure demonstrated the presence of six AAA units within the monomeric molecule[71], calling into question the mechanism of oligomerization and proposed hexameric nature of oligomer RNF213. Here we report that oligomerization of RNF213 on lipid droplets is induced by IFN-I and requires ISGylation of RNF213 itself, an observation that fits with our previous identification of RNF213 as the most ISGylated protein in liver of *Listeria*-infected mice[61]. Various inducers of RNF213 ISGylation could thus potentially facilitate future structural studies on oligomeric RNF213. However, the mechanism of IFN-I-induced oligomerization of RNF213 remains an open question. Relevant in this regard is that RNF213 was reported as a substrate of Protein tyrosine phosphatase 1B (PTP1B or PTPN1), which controls nonmitochondrial oxygen consumption in response to hypoxia[78]. Since PTP1B also affects JAK-STAT signaling[79,80], it would be worthwhile to test whether PTPB1 also regulates RNF213 oligomerization in response to interferon.

It is likely that further processing of ISGylated proteins bound to RNF213 is closely linked to the emerging function of LDs beyond passive lipid storage and lipid homeostasis. LDs have a role in the coordination of immune responses as they participate in the production of pro-inflammatory mediators[81]. LDs have been associated with the IFN-I response and antigen cross presentation[82]. Moreover, an increasing number of studies show that viral and bacterial pathogens target LDs during infection either for nutritional purposes or as part of an anti-immunity strategy[81]. Recent work showed that host cells exploit LD targeting by pathogens by loading LDs with antimicrobial proteins as an intracellular first line of defense[66]. In this study, RNF213 was found to be enriched on LDs isolated from the liver of LPS-treated mice, similar to our results on IFN-treated cells. This is particularly interesting since ISGylation is also induced by LPS[7]. The localization of RNF213 on the surface of LDs could represent a host-cell strategy for fighting intracellular pathogens that hijack LDs. Given that LDs are also sites of viral assembly, RNF213 could interrupt this process to decrease infectivity in the cell. The broad antimicrobial activity of RNF213 that we observed bolsters this hypothesis.

The localization of RNF213 as a general virus restriction factor on the surface of LDs could be further rationalized in the context of enteroviruses, such as coxsackievirus, which replicate at specialized membranous domains named replication organelles[83]. However, whether RNF213 also resides in such lipid-rich replication organelles remains to be determined. On the other hand, ISG15 contributes to the IFN-dependent anti-RSV effect, whereas no role for RNF213 has been reported in the control of RSV. In this context, it is important to note that RSV encodes two prominent nonstructural proteins that profoundly suppress type I IFN induction and signaling[84]. The latter is also true for HSV-1, which has evolved multiple strategies to evade the host antiviral response and to establish latent infections, so far without any known involvement of RNF213[85]. With all three tested viral pathogens we observed a clear, but subtle antiviral effect of RNF213 in vitro, in line with the relatively modest phenotype observed in vivo in RSV infected RNF213 KO mice. Knockdown or overexpression of RNF213 changed viral infection levels two to five fold, in contrast to Listeria where we observed up to 100,000 fold differences in bacterial load, especially in vivo, indicating a more pronounced antibacterial effect. Interestingly, while the protective effect of RNF213 against Listeria was dependent on ISG15, this was not the case for HSV-1, suggesting that different mechanisms underlie the antiviral and antibacterial effects. Of note, a recent study also uncovered that loss of RNF213 increases susceptibility to Rift Valley fever virus infection[62], indicating that the antiviral activity of RNF213 extends beyond the viruses tested here.

Genome-wide siRNA screens in Listeria-infected HeLa cells previously ranked RNF213 as a protective host protein in vitro, in line with our observations[86]. Importantly, the protective effect of RNF213 was lost in ISG15 KO cells, indicating that the antimicrobial activity of RNF213 requires ISG15. Inversely, ISG15 was still functional in RNF213-deficient cells, pointing towards other unknown conjugation-dependent or -independent mechanisms by which ISG15 counteracts Listeria, as shown for many other pathogens (Perng and Lenschow, 2018). Our infection experiments with Listeria in mice, which lack RNF213 corroborated the protective effect in vivo, showing a multi-log increase in the number of colony-forming units in liver and spleen over time compared to WT animals. This increase is larger than what was reported under the same conditions for Isg15-deficient animals[6] and is reminiscent of Listeria infections in mice deficient in key immune signaling pathways necessary to control primary infection such as IFN-gamma[87,88]. Further characterization of the immune response in the Rnf213-deficient animals is both timely and relevant to assess whether a defect in cell-autonomous immunity is underlying the increased susceptibility, as suggested by our in vitro data.

Colocalization of RNF213 with intracellular Listeria further supports a function of RNF213 as a restrictive host factor. Proteins that colocalize with intracellular bacteria are often involved in an antibacterial strategy known as xenophagy. Upon activation, specific E3 ubiquitin ligases, such as LRSAM1[89], SMURF1[90], LUBAC[91,92], or Parkin[93] decorate invading pathogens with ubiquitin chains. Cytosolic autophagy receptors such as p62 or NDP52 recognize these chains and recruit LC3-II to capture the ubiquitinated bacteria into expanding autophagosomes[91]. Since RNF213 has an active E3 ubiquitin ligase module, it is tempting to speculate that RNF213 conjugates ubiquitin chains to invading Listeria and facilitates its capture into autophagosomes. While the present paper was in review, Otten et al. reported that RNF213 indeed acts as a bacterial E3 ligase that can ubiquitinate cytosolic Salmonella, directly on the lipid A moiety of LPS, and that this is required to attract ubiquitin-dependent autophagy receptors and induce antibacterial autophagy[65]. It remains to be determined if RNF213 also ubiquitinates Listeria and on which cell wall components (as a Gram-positive bacterium Listeria lacks LPS). A

vacuole-contained mutant of Listeria did not colocalize with RNF213 and, together with occasional co-staining of RNF213 with ubiquitin on the surface of wild-type bacteria this suggests that RNF213 might target Listeria for autophagy similar to Salmonella. Future studies should investigate the fate of RNF213-decorated Listeria in more depth, especially since in contrast to Salmonella, cytosolic Listeria can evade xenophagy through various mechanisms involving major virulence factors such as LLO, PlcA, and ActA[67–70]. ATPase- and ubiquitin-deficient mutants of RNF213 will allow to assess the implication of both enzymatic activities of RNF213 as well as the role of RNF213 oligomerization. In this regard, loss of bacterial and LD colocalization of the RNF213ΔC mutant could be analogous to what was shown for variants lacking a functional ATPase core, i.e. that ubiquitination activity seems to be required for oligomerization and organelle targeting of RNF213[57]. Taken together, the data presented here provides evidence for colocalization of RNF213 with intracellular Listeria, classifying RNF213 as a pathogen-directed E3 ubiquitin ligase.

Finally and most notably, the largest amount of information regarding RNF213 relates to MMD characterized by progressive stenosis of the internal carotid arteries and the secondary formation of a hazy network of basal collateral vessels in the brain[94,95]. Allelic variations in the RNF213 gene are the most important genetic risk factor to develop MMD; however, the functional role of RNF213 in MMD pathogenesis remains elusive[96]. The best-characterized risk allele is the Asian-specific RNF213 founder variant (p.R4810K), which dramatically increases the risk to develop MMD (e.g., >100-fold in Japan) and likely explains the higher MMD incidence in East-Asia. However, p.R4810K displays a strongly reduced penetrance since the incidence rate of MMD (e.g., 1/10,000 in the Japanese population) is much lower than the population p.R4810K carrier frequency (e.g., 1/50 Japanese carry the RNF213 R4810K risk allele)[96]. This strongly suggests that additional genetic and/or environmental stimuli are required to induce MMD pathogenesis. Our study highlights that RNF213 plays a fundamental role as an antimicrobial host defense effector which could be inappropriately triggered during autoimmune responses, prior infection and/or inflammation in MMD patients[97]. Indeed, previous work suggests that RNF213, like ISG15, is induced by IFN-I and pro-inflammatory cytokines[98]. Here, we show that RNF213 oligomerization could depend on ISGylation of RNF213 and that the protein protects against infection with various pathogens. These data strongly support a role for infectious diseases as trigger to induce MMD in patients carrying RNF213 polymorphisms. It is conceivable that those patients have an impaired immune response to infection that could drive vascular fragility and increased vulnerability of vessels to hemodynamic stress and secondary insults. Together, the data presented here argues for a role of the immune response to infection in the development of MMD and should be fully explored in future studies of the etiology of this mysterious disease.

## Methods

**Antibodies**. The following primary antibodies were used for immunoblotting: mouse monoclonal anti-ISG15 (1:1,000; F-9, sc-166755, Santa Cruz Biotechnology), rabbit monoclonal anti-ISG15 (1:1,000; Abcam [EPR3446] ab133346;), mouse monoclonal anti-ubiquitin (1:1,000; P4D1, #sc-8017, Santa Cruz Biotechnology), mouse monoclonal anti-α-tubulin (1:1,000; B-7, #sc-5286, Santa Cruz Biotechnology), mouse monoclonal anti-GST (1:1,000; B-14, #sc-138, Santa Cruz Biotechnology), rabbit polyclonal anti-GAPDH (1:1,000; FL-335, #sc-25778, Santa Cruz Biotechnology), rabbit polyclonal anti-HA-tag (1:1,000; #H6908, Merck), rabbit polyclonal anti-RNF213 (1:1,000; **#HPA003347**, Merck), mouse monoclonal anti-RNF213 (1:1,000; clone 5C12; Santa Cruz Biotechnology), mouse monoclonal anti-FLAG-tag (1:5,000; M2, #F3165, Merck), rabbit polyclonal anti-LC3B (1:1,000; #PA146286, Thermo Fisher Scientific), rabbit polyclonal anti-tubulin-α (1:1,000; #ab18251, Abcam), anti-p24 GAG (1:1,000; #ab9071, Abcam), rabbit polyclonal

anti-PLIN1 (1:1,000; #ab3526, Abcam), rabbit anti-PLIN2 (1:1,000; #ab108323, Abcam), rabbit polyclonal anti-Rab18 (1:1,000; #ab119900, Abcam), rabbit polyclonal anti-AUP1 (1:1000; #ab224242, Abcam), rabbit polyclonal anti-BiP (1:1,000; Abcam ab21685), rabbit polyclonal anti-Calnexin (1:1,000; #ab10286, Abcam), rabbit polyclonal anti-ATGL (1:1,000; Cell Signaling Technology #2138), goat polyclonal antibody anti-Ribophorin I (1:1,000; #sc-12164, Santa Cruz Biotechnology), mouse monoclonal anti-STAT1 (1:1,000; C-136, #sc-464, Santa Cruz Biotechnology), rabbit monoclonal anti-IFIT1 (1:1,000; #14769, Cell Signaling Technology), rabbit monoclonal phospho-STAT1 (Tyr701) (1:1,000; 58D6, #9167, Cell Signaling Technology), rabbit polyclonal anti-MxA (1:5,000; #H00004599-D01P, Novus Biologicals), rabbit polyclonal anti-MXB (1:1,000; #HPA030235, human protein atlas), and rabbit polyclonal anti-OAS3 (1:1,000; #41440, Cell Signaling Technology). *Listeria* EF-Tu was immunoblotted with rabbit polyclonal antisera raised against α-EF-Tu[99]. Herpes Simplex virus 1 VP5 was immunoblotted with rabbit polyclonal anti-NC-1 antiserum specific for HSV-1 VP5[100]. Respiratory syncytial virus RSV-G was immunoblotted with commercially available goat polyclonal anti-RSV serum (1:1,000; #AB1128, Merck). Coxsackievirus VP1 was immunoblotted with commercially available mouse monoclonal anti-VP1 (1:1,000; 3A8, Mediagnost). Aforementioned primary antibodies were revealed using goat polyclonal anti-mouse-IgG (1:5,000; IRDye® 800CW, Li-COR), goat polyclonal anti-rabbit-IgG (1:5,000; IRDye® 800CW, Li-COR), goat polyclonal anti-mouse-IgG (1:5,000; IRDye® 680RD, Li-COR), or goat polyclonal anti-rabbit-IgG (1:5,000; IRDye® 680RD, Li-COR), except for anti-RSV serum which was revealed with secondary anti-goat (1:1,000; #sc-2020, Santa Cruz biotechnology). For microscopy, cells were stained with an anti-ubiquitin FK2 antibody (1:100; #ST1200, Merck) with Goat anti-mouse IgG (H + L) Superclonal™ Recombinant Secondary Antibody, Alexa Fluor 647 (1:1,000; #A28181, Thermo Fisher Scientific). To visualize intracellular *Listeria*, cells were stained with anti-R11 (1:200; gift from the Cossart laboratory, raised in house[101]) and Goat anti-rabbit IgG (H + L) Superclonal™ Recombinant Secondary Antibody, Alexa Fluor 647 (1:1,000; #A27040, Thermo Fisher Scientific). Actin was visualized with Phalloidin AlexaFluor 350 Stain (1:40 in methanol; #A22281, Thermo Fisher Scientific).

**Cell culture.** Hek293T cells originate from[102] while HeLa cells (ATCC® CCL-2™), A549 cells (ATCC® CCL-185™) THP-1 cells (ATCC® TIB-202™), and primary human CD14 + monocytes (ATCC® PCS-800-010™) were purchased from ATCC. HEK293T and A549 cells were cultured in DMEM medium (#31966047, Thermo Fisher Scientific) supplemented with 10% fetal bovine serum (FBS, #10270106, Thermo Fisher Scientific). HeLa cells were maintained in MEM medium (#M2279, Merck) supplemented with 10% FBS, 1% glutamax (#35050038, Thermo Fisher Scientific), 1% nonessential amino acids (#11140035, Thermo Fisher Scientific), 1% sodium pyruvate (#11360039, Thermo Fisher Scientific), and 1% hepes (#15630056, Thermo Fisher Scientific). THP-1 cells were maintained in RPMI 1640 medium (#61870044, Thermo Fisher Scientific) supplemented with 10% FBS and 2 mM L-glutamine (#25030024, Thermo Fisher Scientific). THP-1 cells were differentiated to macrophages by complementing the media with 10 ng/ml phorbol 12-myristate 13-acetate (#P8139, Merck) for 7 days, followed by rest for 24 h in medium RPMI 1640 medium supplemented with 10% FBS, 2 mM L-glutamine. Primary human CD14+ monocytes were maintained in Hank's Balanced Salt Solution (HBSS, #14175095, Thermo Fisher Scientific) supplemented with 10% FBS. HEK293T, HeLa, and A549 cells were authenticated by Eurofins.

**Strain generation.** Strains used in this study are listed in Supplementary Table 1. To generate mCherry EGD we codon optimized mCherry for *Listeria monocytogenes* using the JCAT website[103], ordered the codon optimized geneblock from IDT and subcloned mCherry into pAD, electroporated competent *Listeria monocytogenes* EGD (BUG600) and selected for chromosomal integration of the plasmid.

**Plasmid transfection.** Plasmid transfection was performed with Polyethylenimine (PEI, #23966-1, Polysciences) as transfection reagent at a ratio PEI/cDNA of 5:1 (w/w). Plasmids were used at a final concentration of 1 μg DNA/10$^6$ Hek293T cells or 1 μg DNA/5 × 10$^5$ HeLa cells. The following plasmid were used: pMD2.G (VSV-G), pcDNA3-FLAG-VSV-G, pMET7-GAG-EGFP, pMET7-GAG-ISG15GG, pMET7-GAG-ISG15AA, pMET7-GAG-ISG15 precursor, pGL4.14-3xFLAG-RNF213, pGL4.14-3xFLAG-RNF213ΔC, pGL4.14-3xFLAG-RNF213$_{R4810K}$, pMet7-FLAG-C-domain-RNF213, pRK5-HA-RNF4, pCMV3-FLAG-HOIP (RNF31), pMET7-Flag-eGFP, pcDNA3.1-HA-ISG15GG, pccDNA3.1-HA-ISG15AA, pSVsport (Mock plasmid), pcDNA3.1-hUbe1L (E1), pcDNA3.1-Ubch8 (E2) and pTriEx2-hHERC5 (E3), pDEST-eGFP-RNF213, and pDEST-eGFP-RNF213ΔC. All aforementioned plasmids are available on addgene.org or listed in Supplementary Table 2. All used primers are listed in Supplementary Table 3.

**SiRNA transfection.** A commercially available siRNA pool was used to knockdown human RNF213 (#M-023324-02, GE Healthcare Dharmacon) in Figs. 2, 3, 4, 5, 6, and 7. As control siRNA treatment, a nontargeting scramble siRNA (#D-001210-01-05, GE Healthcare Dharmacon) was used in the experiments shown in Fig. 3 and Supplementary Fig. 6c while a pool of four scrambled siRNAs (#D-001206-13-05, GE Healthcare Dharmacon) was used in the experiments

shown in Figs. 2, 4, 5, 6, and 7. In all experiments, siRNAs were transfected with DharmaFECT transfection reagent (#T-2001-02, GE Healthcare Dharmacon) according to the instructions of the manufacturer. For Supplementary Fig. 9b, the siRNA D-023324-05 (GE Healthcare Dharmacon) and Stealth RNAis (#HSS126645, #HSS184009, Thermo Fisher Scientific) were also transfected in HeLa cells to knockdown RNF213 with DharmaFECT and Lipofectamine 3000 transfection reagent (#L3000008, Thermo Fisher scientific), respectively. For Fig. 7, a commercially available siRNA pool was used to knockdown ISG15 (#M-004235-04-005, GE Healthcare Dharmacon). In these experiments, a reverse siRNA transfection protocol was adopted to knockdown the expression of ISG15 and RNF213 genes in HeLa cells prior to *Listeria* infection. A single siRNA, custom produced by Dharmacon, was used to knockdown RSV-N expression as described in ref. [104]. Immunoblotting assays were conducted to confirm reduction of protein expression levels. The aforementioned siRNAs are listed in Supplementary Table 3.

**Generation of knockout cell lines.** The RNF213 knockout HeLa cell line was generated by using a CRISPR/Cas9 approach. Target sequences were selected by CRISPOR (CRISPOR.org,[105]). Oligonucleotides (5'CACCGGAGGCAGCCTCTCT CCGCAC and 5'AAACGTGCGGAGAGAGGCTGCCTCC; 5'CACCGTGCAGC CCCCATAGCAGGTG and 5'AAACCACCTGCTATGGGGGCTGCAC) were synthesized by ID&T (Leuven, Belgium) and cloned into the pSpCas9(BB)-2A-Puro plasmid (pXP459, Addgene #48139). HeLa cells were co-transfected with the two RNF213-targeting plasmids with Lipofectamine 3000 (L3000008, Thermo Fisher Scientific) as described above. Cells were selected with 2 μg/mL with Puromycin (#P8833, Merck) for 48 h. Cells were diluted and plated in 96-well plate for single-clone selection. The absence of RNF213 protein expression was confirmed by immunoblotting and the absence of the highest-scoring predicted off-target effect in a coding region (in MLK1) was confirmed by genomic PCR and next-generation sequencing of the amplicon. The ISG15 knockout HeLa cell line was generated as previously reported[12]. Briefly, using a CRISPR/Cas9 approach, single-guide RNAs (sgRNAs) targeting ISG15 were designed using CRISPOR (CRISPOR.tefor.net): 5'-CACCGGAACTCATCTTTGCCAGTACAGG-3' and 5'-AAAC GTACTGGCAAAGATGAGTTCC-3'. The single-strand gRNA sequences were duplexed and ligated into the Bsm BI–digested and dephosphorylated lenti-CRISPRv2 vector. Using the lentiviral packaging plasmids psPAX2 (Addgene) and pMD2.G (Addgene), HeLa cells were transduced and cultured for 72–96 h prior selection using puromycin treatment at a concentration of 1.5 μg/mL for 24–48 h. Single-clone selection took place after puromycin treatment by dilution. ISG15 deficiency was tested by IFN-β stimulation and subsequent western blotting for ISG15. Control cells were generated by transduction of lentiviral vectors containing the Cas9 sequence without gRNAs. The aforementioned guide RNAs are listed in Supplementary Table 3.

**SDS-PAGE and immunoblotting.** Cells were lysed in 2× Laëmmli buffer containing 125 mM Tris-HCl pH 6.8, 4% SDS, 20% glycerol, 0,004% Bromophenol blue supplemented with 20 mM DTT. Protein samples were boiled for 5 min at 95 °C and sonicated prior to SDS-PAGE. Samples were loaded on 4–20% polyacrylamide gradient gels (#M42015, Genescript), 4–15% Mini-PROTEAN TGX Gels (#4561084, Biorad), 3–8% Criterion XT tris-acetate gel (#3450130, Biorad), or 4–15% Criterion TGX gel (#5671083, Biorad) according to the guidelines of the manufacturer. For detection of RNF213, proteins were separated on a 3–8% Criterion XT tris-acetate gel (Biorad) or 4–15% Criterion TGX gel according to the instructions of the manufacturer. Proteins were transferred to PVDF membrane (#IPFL00010, Merck) for 3 h at 60 V with Tris/Boric buffer at 50 mM/50 mM. Membranes were blocked for 1 h at room temperature (RT) with blocking buffer (#927-50000, LI-COR) and incubated with primary antibodies overnight at 4 °C diluted to 1:1000 in TBS. The next day, membranes were washed three times for 15 min with TBS-Tween 0.1% (v/v) buffer and further incubated at RT for 1 h with the appropriate secondary antibody. Membranes were washed twice with TBS-tween 0.1% (TBS-T) and once with TBS prior to detection. Immunoreactive bands were visualized on a LI-COR-Odyssey infrared scanner (Li-COR).

**Lipid droplets isolation and cellular fractionation.** THP-1 cells treated or not with interferon-β (100 U/mL; 8-10 h, #11343524, Immunotools) were washed and scraped in ice-cold hypotonic disruption buffer containing 20 mM tricine and 250 mM sucrose pH 7.8, 0.2 mM PMSF followed by homogenization in a chilled glass homogenizer. THP-1 lysates underwent a nitrogen bomb at a pressure of 35 bar for 15 min on ice to complete cellular disruption. Lysates were centrifuged at 3000 × *g* for 10 min at 4 °C and the post-nuclear supernatants was loaded at the bottom of 13 mL centrifuge tubes and overlaid with ice-cold wash buffer containing 20 mM HEPES, 100 mM KCl and 2 mM MgCl2 at pH 7.4. Tubes were centrifuged in a swing-out rotor at 182,000 × *g* at 4 °C for 1 h. Lipid droplets were collected from the top of the gradient. Lipid droplet-enriched samples were centrifuged at 20,000 × *g* for 5 min at 4 °C to separate the floating lipid droplets from the aqueous fraction. Lipid droplets were washed three times by adding 200 μL ice-cold wash buffer followed by centrifugation at 20,000 × *g* for 5 min at 4 °C. Lipid droplets were resuspended in 100 μL 2× Laëmmli buffer and further processed for immunoblot analysis as described above. To isolate the membrane fraction, THP-1 lysates were centrifuged at 3000 × *g* for 10 min at 4 °C and the pellet was washed

three times by resuspension with 1 mL of ice-cold wash buffer followed by centrifugation. The insoluble fractions were resuspended in 2× Laëmmli buffer and further processed for immunoblot analysis as described above. To isolate the cytosolic fraction, THP-1 lysates were centrifuged at $3000 \times g$ for 10 min and 1 mL aliquot was sampled from the middle of the lysate. The 1 mL aliquot was further centrifuged at $270,000 \times g$ for 1 h at 4 °C in a microcentrifuge tube using a TLA100.3 rotor. The supernatants were collected and mixed to a final 2× Laëmmli buffer prior further processing for immunoblot analysis as described above. For cellular fractionation in HeLa cells, cells were grown on 100 cm² petri dishes and either untreated or treated for 24 h with interferon-β at 10 ng/ml (#11343524, Immunotools). Cells were processed as described in[106]. Briefly, cells were detached with trypsin, washed with PBS and lysed in a buffer containing 300 mM sucrose, 5 mM Tris-HCl, 0.1 mM EDTA, pH 7.4 and a protease inhibitor cocktail (Roche, cOmplete, Mini, EDTA-free tablet) followed by homogenization in a chilled glass homogenizer. Lysates were centrifuged at $800 \times g$ for 8 min at 4 °C followed by separation of the soluble and membrane fractions by a second centrifugation step at $20,000 \times g$ for 120 min at 4 °C. The membrane-associated fraction was redissolved in lysis buffer to a volume equal to the volume of the soluble fraction. Samples were mixed with Laëmmli buffer and processed for immunoblotting as described above.

**Glycerol gradient analyses**. Lysates prepared as described for the purification of lipid droplets were loaded onto a glycerol gradient (10–40% (w/v)). The gradients were centrifuged at $237,000 \times g$ for 20 h at 4 °C, using a SW 55 Ti rotor and Beckman L-80 ultracentrifuge. Twenty fractions for each sample were collected in 2 mm increments using a BioComp Piston Gradient fractionator. Fractions were concentrated by TCA precipitation and analyzed by immunoblotting as described above. For immunoprecipitations, each gradient fraction was first desalted over Amicon filter to remove glycerol and resuspended in 1 ml of resuspension buffer containing 20 mM HEPES, 100 mM KCl, and 2 mM MgCl2 at pH 7.4. Resuspended fractions were precleared with Protein G-agarose beads by incubation for 1 h at 4 °C. Immunoprecipitation of RNF213 was performed from precleared fractions by incubation with RNF213 antibody for 3 h at 4 °C with end-over-end rotation. Beads were washed twice in 1 mL of resuspension buffer by centrifugation and samples were eluted into 1× Laëmmli buffer by heating at 60 °C and processed for immunoblotting as described above.

**In vitro infection with _Listeria monocytogenes_**. _Listeria monocytogenes_ (EGD BUG600 strain) was grown in brain heart infusion (BHI) medium at 37 °C. _Listeria_ were cultured overnight and then subcultured 1:10 in BHI medium for 2 h at 37 °C. Bacteria were washed three times in PBS and resuspended in medium without FBS prior to infection. HeLa cells were grown in 6-well plates and infected with _Listeria_ at a multiplicity of infection (MOI) of 25. Right after infection, plates were centrifuged at $1000 \times g$ for 1 min followed by incubation for 1 h at 37 °C to allow entry of the bacteria. Afterwards, cells were washed two times with PBS and then grown in MEM medium with 10% FBS, 1% glutamax, 1% nonessential amino acids, 1% sodium pyruvate, 1% hepes, supplemented with 40 µg/mL of gentamicin to kill extracellular bacteria. For immunoblotting, infected cells were washed with ice-cold PBS, lysed in 2× Laëmmli buffer and further processed as described above. To count the number of intracellular bacteria, HeLa cells were washed and lysed with miliQ water to release intracellular bacteria. Colony Forming Units (CFUs) were determined by serial dilution and plating on BHI agar.

**In vivo infection with _Listeria monocytogenes_**. _Listeria monocytogenes_ (EGD BUG600 strain) was grown as described above. Female and Male C57BL/6 mice (_Rnf213⁺/⁺_ or _Rnf213⁻/⁻_) between 8 and 12 weeks of age were infected intravenously by tail vein injection with $5 \times 10^5$ bacteria per animal. Mice were sacrificed 72 h following infection. CFUs per organ (liver or spleen) were enumerated by serial dilutions after tissue dissociation in sterile saline. The animals were housed in a temperature- (21 °C) and humidity- (60%) controlled environment with 12 h light/dark cycles; food and water were provided ad libitum. The animal facility operates under the Flemish Government License Number LA1400536. All experiments were done under conditions specified by law and authorized by the UGent Institutional Ethical Committee on Experimental Animals.

**In vitro infection with Herpes Simplex Virus Type 1 strain C12 (HSV-1 C12)**. HSV-1 C12 is a recombinant HSV-1 strain SC16 containing a cytomegalovirus (HCMV) IE1 enhancer/promoter-driven enhanced green fluorescent protein (EGFP) expression cassette in the US5 locus (kindly provided by Stacey Efstathiou (University of Cambridge, Cambridge, UK). HSV-1 C12 was propagated in Vero cells and titrated by plaque assay with Vero cells according to Grosche et al.[107]. HeLa cells were grown in 96-well plates and infected with HSV-1 C12 at MOI 0.1 (Fig. 4a–c) or MOI 0.05 (Fig. 4d, e). Cells were treated either without or with 1000 U/mL IFNα2a (#11343504, Immunotools). Cells were maintained in 2% FBS FluoroBrite DMEM Media. GFP fluorescence was monitored at 24 h intervals using the Infinite® 200 PRO Reader and Tecan iControl Software (Tecan Life Sciences). Each individual measurement ($n = 4$) was normalized to the mean value of the uninfected wells ($n = 4$). The relative area under the curve from 0 to 72 h post infection was calculated for each growth curve according to[64]. For immunoblot

analysis, cells 24 h p.i were washed with ice-cold PBS, lysed in 2× Laëmmli buffer and further processed as described above.

**In vitro infection with coxsackievirus B3 (CVB3)**. CVB3 Nancy strain was propagated in Green Monkey Kidney Cells and quantified by plaque assay in HeLa cells as described below. HeLa cells were grown in six-well plates and infected with CVB3 (MOI 0.01) in serum-free medium. After 1 h, the CVB3 inoculum was replaced and HeLa cells were cultured with MEM medium with 10% FBS, 1% glutamax, 1% nonessential amino acids, 1% sodium pyruvate, and 1% hepes. For immunoblotting, the cells were washed with ice-cold PBS, lysed in 2× Laëmmli buffer and further processed as described above to reveal VP1. For relative quantification of VP1, the intensity of the VP1 band was divided by the intensity of the GAPDH band, as loading control, and normalized to the intensity of the VP1 band in the control condition. For qRT-PCR, cells were lysed in 500 µl TRIzol™ Reagent. RNA was extracted from the aqueous phase after chloroform addition and precipitated by addition of isopropyl alcohol. Two hundred and fifty nanograms RNA per sample was used for reverse transcription to cDNA. For relative quantification, the ΔΔCt method was used with Hypoxanthine Phosphoribosyl transferase (HPRT) as housekeeping gene. Plaque-forming units (PFU)/mL were determined by serial dilution on confluent HeLa cells.

**In vitro infection with respiratory syncytial virus (RSV)**. RSV-A2, an A subtype of RSV (ATCC, VR-1540, Rockville), was propagated in HEp-2 cells and quantified by plaque assay with A549 cells as described below. A549 cells were grown in 6-, 48-well, or 96-well plates, respectively, for immunoblot, titration or plaque assay. A549 cells were infected in serum-free medium with RSV-A2 at MOI 0.001, MOI 0.02 or MOI 0.005 for the respective experiment. After 4 h, the RSV inoculum was replaced and A549 cells were cultured in DMEM medium with 10% FBS. For immunoblot analysis, cells were washed with ice-cold PBS, lysed in 2× Laëmmli buffer and further processed as described above. In the plaque and titration assay, A549 cells were treated either without or with 10 ng/mL interferon-β (#11343524, Immunotools) for 48 h. For plaque assays, the DMEM medium with 10% FBS was supplemented with 0,6% (w:v) avicel RC-851 (FMC biopolymers) and the cells were fixed 6 days post RSV infection with 4% paraformaldehyde (PFA) in PBS and permeabilized with 0,2% Triton-X100 in PBS. RSV plaques were stained with a polyclonal goat anti-RSV serum and secondary anti-goat IgG. Plaques were visualized by TrueBlue™ Peroxidase substrate (5510, Sera Care). Each plaque was quantified in Fiji ImageJ 1.51n. In titration assays, Plaque Forming Units (PFU)/mL were determined by serial dilution on confluent A549 cells. Supernatants were collected and mixed (1:1) with a 40% sucrose solution in HBSS, snap frozen and stored at −80 °C.

**In vivo infection with respiratory syncytial virus (RSV)**. An in-house generated mouse-adapted RSV-A2[108] was used for in vivo infection. RSV was propagated and quantified by plaque assay in HEp-2 cells as described above. Twelve-week-old C57Bl/6 J mice (_Rnf213⁺/⁺_ and homozygous _Rnf213⁻/⁻_ littermates) were infected intranasally with $6 \times 10^6$ PFU after isoflurane-based sedation. Seventeen mice (9 wild-type (7 females, 2 males) and 8 knockout (5 females, 3 males) littermates were euthanized at day 5 post infection to determine the lung viral titers by serial dilution after tissue dissociation in sterile HBSS supplemented with 20% sucrose (Supplementary Fig. 12e). Ten mice (5 wild-type and 5 knockout littermates, 2 males and 3 females each) were used to follow-up bodyweight (Supplementary Fig. 12f) and survival rate (Supplementary Fig. 12g) for 14 days post infection. The animals were housed in a temperature- (21 °C) and humidity- (60%) controlled environment with 12 h light/dark cycles; food and water were provided ad libitum. The animal facility operates under the Flemish Government License Number LA1400536. All experiments were done under conditions specified by law and authorized by the UGent Institutional Ethical Committee on Experimental Animals.

**Isolation and cell culture of BMDM's**. Female and Male C57BL/6 mice (_Rnf213⁺/⁺_ or _Rnf213⁻/⁻_) between 8 and 12 weeks of age were used to isolate bone marrow cells from femurs. Isolated bone marrow cells were cultured for 7 days at a density of $10^5$ cells/mL in DMEM/F-12 medium, supplemented with 10% FBS, 10 units/ml penicillin, 10 µg/mL streptomycin, and 20 ng/mL murine M-CSF (#PMC2044, Thermo Fisher Scientific) at 37 °C in a humidified atmosphere with 5% $CO_2$. On day 3, fresh medium containing 40 ng/ml M-CSF was added and cells were further differentiated for 4 days.

**Microscopy analysis of interferon-treated BMDM's**. The day prior to an experiment, BMDMs were seeded in IBIDI µ-Slide eight-well chambers (#80826, Ibidi) and treated with oleic acid conjugated to BSA (molar ratio 5:1 in PBS) at a concentration of 200 µM, if mentioned. 24 h prior fixation, cells were treated with interferon-β (#11343524, Immunotools) at 10 ng/mL or left untreated. BMDMs were fixed in PBS with 4% PFA for 20 min at RT. LHCS LipidTOX™ Red Neutral Lipid Stain (#H34477, Thermo Fisher Scientific) and Hoechst (#H1399, Thermo Fisher Scientific) were used to visualize lipid droplets and nuclei respectively. Cells were mounted using Prolong Diamond Mounting Medium (#P36965, Thermo Fisher Scientific). Images were acquired using a confocal laser scanning microscope

(LSM880 with Airyscan, Carl Zeiss Microscopy) and the software ZEN 2.3 Pro. Images were deconvoluted using Zen 3.1 (Blue Edition) for optimal resolution. The number and relative area of LDs were analyzed using Volocity 6.0 software.

**Microscopy analysis of *Listeria monocytogenes* infected HeLa cells.** Overnight cultures of *Listeria monocytogenes* were diluted in Brain Heart Infusion media (BD) and grown to exponential phase (OD 0.8–1), washed three times in serum-free MEM (Life Technologies), and resuspended in serum-free MEM at corresponding MOIs. A fixed volume was then added to each well. Cells were centrifuged for 1 min at $201 \times g$ to synchronize infection. The cells were then incubated with the bacteria for 1 h at 37 °C, 5% CO$_2$. Following this incubation, the cells were washed at room temperature with 1× DPBS, and then fed with full growth medium (with 10% fetal bovine serum) supplemented with 20 µg/ml gentamicin to kill extra-cellular bacteria.

HeLa cells were plated on coverslips in six-well dishes the day prior to an experiment and transfected with 2.0 µg of EGFP-RNF213 or eGFP-RNF213ΔC at the time of seeding with Fugene® HD (Promega) at a ratio of 3 uL reagent: 1 ug DNA. *Listeria monocytogenes* EGDe prfA* at MOI 1, EGDe prfA*ΔplcAΔplcBΔhly at MOI 75, or mCherry EGD at MOI 20 for 6 h, respectively. For experiments with eGFP-RNF213 and eGFP-RNF213ΔC, mCherry EGD infection of cells was performed at MOI 10 for 18 h (unless otherwise indicated). After infection, cells were subsequently fixed with 4% PFA (Electron Microscopy Sciences, Hatfield, Pennsylvania) in PBS for 20 min at room temperature and permeabilized with 0.5% Triton in PBS for 10 min at room temperature. Cells were then counterstained for *Listeria* (R11 anti-*Listeria*, raised in house) with goat anti-rabbit IgG (H + L), Superclonal™ Recombinant Secondary Antibody, Alexa Fluor 647 or ubiquitinated proteins (FK2, Millipore sigma) with goat anti-mouse IgG (H + L), Superclonal™ Recombinant Secondary Antibody, Alexa Fluor 647. Coverslips were mounted using ProLong Diamond Antifade Mountant with or without 4′,6-diamidino-2-phenylindole (DAPI, Thermo Fisher Scientific). Images were acquired using an inverted wide-field fluorescence microscope (Axio Observer 7, Carl Zeiss Microscopy, Germany) equipped with an Axiocam 506 mono camera and the software ZEN 2.3 Pro. Where indicated images were deconvolved using Zen 3.1 (Blue Edition) for optimal resolution using default deconvolution settings (better, fast, regularized inverse filter).

**Virotrap sample preparation for LC-MS/MS and immunoblotting analysis.** Virotrap experiments were described as described in ref. [109]. Briefly, the day before transfection $10^6$ Hek293T cells (authenticated) were seeded in four T75 flasks for each condition. Cells were then transfected with four different bait proteins, each fused via their N-terminus to HIV-1 GAG: mature ISG15 (ending on -LRLRGG), mature nonconjugatable ISG15 (ending on -LRLRAA), full-length ISG15 precursor, and eDHFR (dihydrofolate reductase from *Escherichia coli*) as control. VSV-G and FLAG-VSV-G were co-transfected to allow single-step purification of the produced particles. The day after transfection, these cells were treated with interferon-α for 24 h (10 ng/mL; #11343596, Immunotools) during the particle production phase. Supernatant was harvested 40 h after transfection followed by centrifugation and filtration (0.45 µM; SLHV033RS, Merck Millipore) to remove cellular debris. Virotrap particles containing protein complexes were purified in a single step using biotinylated anti-FLAG BioM2 antibody (#F9291, Merck) and Dynabeads MyOne Streptavidin T1 Beads (#65601, Thermo Fisher Scientific), and were consecutively eluted by competition using FLAG peptide (#F3290, Merck). For immunoblot analysis, samples containing the purified VLPs were mixed 1:1 with 4× Laëmmli buffer complemented with 20 mM DTT further analyzed as described above. After FLAG elution, samples were processed with Amphipols A8-35 (#A835, Anatrace), digested using trypsin (V5111, Promega), and acidified. Each condition (4 baits, ±interferon-α) was analyzed in quadruplicate, leading to a total of 32 samples for LC-MS/MS analysis. Peptides were purified on Omix C18 tips (Agilent), dried and redissolved in 20 µl loading solvent A (0.1% trifluoroacetic acid in water/acetonitrile (ACN) (98:2, v/v)) of which 2.5 µl was injected for LC-MS/MS analysis.

**GST pull-down for LC-MS/MS and immunoblotting.** HEK293T, HeLa or differentiated THP-1 cells were grown at ~80–90% confluence in 15 cm culture dishes (1 petri dish/sample). HEK293T cells were treated with interferon-α while HeLa and THP-1 cells were treated with interferon-β, at 10 ng/mL for 24 h (#11343596, #11343524, Immunotools). Each condition (2 baits, 3 cell lines) was analyzed in triplicate, leading to a total of 18 samples for LC-MS analysis. Cells were washed three times with PBS with Ca$^{2+}$ and Mg$^{2+}$ and scraped in 1 mL lysis buffer containing 50 mM Tris.HCl (pH 8.0), 150 mM NaCl, 1% triton-x-100 (v/v), 1 mM PMSF, and a protease inhibitor cocktail (Roche, cOmplete, Mini, EDTA-free tablet, 4693159001). Samples were incubated 60 min on ice or 20 min under end-over-end agitation at 4 °C. Lysates were cleared by centrifugation for 15 min at $16,000 \times g$ at 4 °C to remove insoluble components. Fifteen microliters of magnetic glutathione beads (Pierce Thermo Fisher Scientific) were incubated with 4 µg of purified GST-tagged ISG15 (R&D systems, UL-600-500) or GSTP1 (#G5663, Merck) under agitation. For immunoblotting, glutathione beads were also decorated with GST-tagged-SUMO1 (R&D systems, UL-710-500) or GST-tagged-Ubiquitin (R&D systems, U-540-01M). Incubation was performed overnight at 4 °C in 500 µL

50 mM Tris.HCl, 150 mM NaCl, 1% triton-x-100 (v/v), 1 mM DTT after which decorated glutathione beads were blocked by addition of BSA to a final concentration of 2% (w/v) and incubation for an additional hour at 4 °C. Then, the cleared lysates were incubated overnight at 4 °C with the decorated glutathione beads under agitation to allow binding of proteins to ISG15. The following day, the beads were precipitated with a magnetic stand, washed once with 1 mL wash buffer containing 50 mM Tris.HCl (pH 8.0), 150 mM NaCl, 1% triton-x-100 (v/v). For immunoblotting, the beads were washed two extra times with 1 mL wash buffer for a total of three washes. Beads were mixed with 60 µL 2× Laëmmli buffer complemented with 20 mM DTT and boiled at 95 °C for 10 min. Beads were precipitated with a magnet and the supernatants were further analyzed by immunoblotting as described above. In the LC-MS/MS analysis, beads were washed three extra times with 1 mL trypsin digestion buffer containing 20 mM Tris-HCl pH 8.0, 2 mM CaCl2. Washed beads were resuspended in 150 µl digestion buffer and incubated for 4 h with 1 µg trypsin (Promega) at 37 °C. Beads were removed, another 1 µg of trypsin was added and proteins were further digested overnight at 37 °C. Peptides were purified on Omix C18 tips (Agilent), dried and redissolved in 20 µl loading solvent A (0.1% trifluoroacetic acid in water/acetonitrile (ACN) (98:2, v/v)) injected for LC-MS/MS analysis. In the experiment with HeLa cells, 15 µL of samples was injected while 5 µL was injected for the experiment with THP-1 and Hek293T cells.

**Immunoprecipitation (IP) for LC-MS/MS and immunoblotting.** HEK293T and HeLa cells were grown at ~80–90% confluence in 15 cm culture dishes (1 petri dish/sample). In the IP-immunoblotting experiments, cells were transfected either with FLAG-tagged-eGFP or FLAG-RNF213 plasmids, plus, the ISG15 conjugation machinery with HA-tagged-ISG15 matured, UBA7, UBCH8, and HERC5 plasmids. In the IP-MS experiment, cells were transfected either with plasmid coding for HA-tagged-ISG15AA or a mock plasmid. HEK293T cells were treated with interferon-α at 24 h and 48 h while HeLa cells were treated with interferon-β at 24 h. Intereferon-α/β treatment were performed at 10 ng/mL (#11343596, #11343524, Immunotools). Each condition (2 baits, 3 types of interferon-treated cells) was analyzed in triplicate, leading to a total of 18 samples for LC-MS/MS analysis. Cells were washed three times with PBS with Ca$^{2+}$ and Mg$^{2+}$ and scraped in 1.5 mL lysis buffer containing 50 mM Tris.HCl (pH 8.0), 150 mM NaCl, 1% triton-x-100 (v/v), 1 mM PMSF and a protease inhibitor cocktail (Roche, cOmplete, Mini, EDTA-free tablet, 4693159001). Samples were incubated 60 min on ice or 20 min under end-over-end agitation at 4 °C. Lysates were cleared by centrifugation for 15 min at $16,000 \times g$ at 4 °C to remove insoluble components. Supernatants were incubated with Pierce anti-HA magnetic beads (#88836, Pierce Thermo Fisher Scientific) or anti-FLAG magnetic beads (#M8823, Merck). Lysates were incubated at 4 °C under agitation overnight. The beads were precipitated with a magnetic stand and subsequently washed once with wash buffer containing 50 mM Tris.HCl (pH 8.0), 150 mM NaCl, 1% triton-x-100 (v/v). In the IP-immunoblotting experiments, the beads were washed two extra times with wash buffer. Beads were mixed with 60 µL 2× Laëmmli buffer and heated at 60 °C for 10 min. Beads were precipitated with a magnet and the supernatants were further analyzed by immunoblot analysis as described above. In the IP-MS experiment, the beads were washed three extra times with trypsin buffer containing: 20 mM Tris-HCl pH 8.0, 2 mM CaCl$_2$. The proteins were on-bead digested with 1 µg trypsin (Promega) for 4 h at 37 °C under agitation. Then, the beads were precipitated and the supernatants were mixed with 1 µg of trypsin and further digested overnight at 37 °C under agitation. Peptides were desalted on reversed phase C18 OMIX tips (Agilent), all according to the manufacturer's protocol. Purified peptides were dried, redissolved in 20 µl loading solvent A (0.1% trifluoroacetic acid in water/acetonitrile (ACN) (98:2, v/v)) and 5 µL were injected for LC-MS/MS analysis.

**LC-MS/MS and data analysis.** LC-MS/MS analysis was performed on an Ultimate 3000 RSLCnano system (Thermo Fisher Scientific) in line connected to a Q Exactive mass spectrometer or a Q Exactive HF (Thermo Fisher Scientific). Trapping was performed at 10 µl/min for 4 min in loading solvent A on a 20 mm trapping column (100 µm internal diameter (I.D.), 5 µm beads, C18 Reprosil-HD, Dr. Maisch, Germany) before the peptides were separated on a 150 mm analytical column packed in the needle (75 µm I.D., 1.9 µm beads, C18 Reprosil-HD, Dr. Maisch). Prior to packing of the column, the fused silica capillary had been equipped with a laser pulled electrospray tip using a P-2000 Laser Based Micropipette Peller (Sutter Instruments). Alternatively, peptides were separated on a 500 mm long micro pillar array column (µPAC™, PharmaFluidics) with C18-endcapped functionality. This column consists of 300 µm wide channels that are filled with 5 µm porous-shell pillars at an inter pillar distance of 2.5 µm. With a depth of 20 µm, this column has a cross section equivalent to an 85 µm inner diameter capillary column. Peptides were eluted from the analytical column by a nonlinear gradient from 2 to 55% solvent B (0.1% FA in water/acetonitrile (2:8, v/v)) over 30, 60, or 90 min at a constant flow rate of 250 or 300 nL/min, followed by a 5 min in 99% solvent B. Then, peptides were eluted by a 5 min in 99% solvent B. The column was then re-equilibrated with 98% solvent A (0.1% FA in water) for 15 min. In the virotrap experiment, the mass spectrometer was operated in positive and data-dependent mode, automatically switching between MS and MS/MS acquisition for the 5, 10, or 12 most abundant ion peaks per MS spectrum. Full-scan MS spectra (400–2000 *m/z*) were

acquired at a resolution of 70,000 (at 200 $m/z$) in the orbitrap analyzer after accumulation to a target value of 3E6 for a maximum of 80 ms. The 10 most intense ions above a threshold value of 1.7E4 were isolated in the trap with an isolation window of 2 Da for maximum 60 ms to a target AGC value of 5E4. Precursor ions with an unassigned, or with a charge state equal to 1, 5–8, or >8 were excluded. Peptide match was set on "preferred" and isotopes were excluded. Dynamic exclusion time was set to 50 s. Fragmentation were performed at a normalized collision energy of 25%. MS/MS spectra were acquired at fixed first mass 140 $m/z$ at a resolution of 17,500 (at 200 $m/z$) in the Orbitrap analyzer. MS/MS spectrum data type was set to centroid. The polydimethylcyclosiloxane background ion at 445.12002 was used for internal calibration (lock mass) in addition to the ion at 361.14660 corresponding to a tri-peptide of asparagine that was spike-in the MS solvents as described in[110]. Similar settings were used for the GST pull-down and immunoprecipitation experiments.

Data analysis was performed with MaxQuant (version 1.6.3.4) using the Andromeda search engine with default search settings including a false discovery rate set at 1% on the PSM, peptide and protein level. All spectral data files were searched with MaxQuant against all human proteins in the Uniprot/Swiss-Prot database (database release version of January 2019 containing 20,413 protein sequences (taxonomy ID 9606), downloaded from www.uniprot.org). For virotrap, the 32 recorded spectral data files were searched together and the search database was complemented with the GAG, VSV-G, and eDHFR protein sequences. For the GST pull-down experiments, three different searches were performed, one search for each cell line. For the IP experiments, three different searches were performed, two searches for HEK293T cells treated with interferon-α for 24 h and 48 h, and one search for HeLa cells treated with interferon-β for 24 h. For GST pull-down and IP experiment, each search comprised six spectral data files. The mass tolerance for precursor and fragment ions was set to 4.5 ppm and 20 ppm, respectively, during the main search. Enzyme specificity was set as C-terminal to arginine and lysine (trypsin), also allowing cleavage at arginine/lysine-proline bonds with a maximum of two missed cleavages. Variable modifications were set to oxidation of methionine (sulfoxides) and acetylation of protein N-termini. Matching between runs was enabled with an alignment time window of 20 min and a matching time window of 1 min. Only proteins with at least one peptide were retained to compile a list of identified proteins. In virotrap, 1214 proteins were identified in all 32 samples. In the GST pull-down, 599 proteins were identified in the experiment with THP-1 cells, 800 with HeLa cells, and 933 with HEK293T cells. In the IP experiments, 401 proteins were identified in the experiment with HEK293T cells treated with interferon-α for 24 h; 3795 proteins were identified in the experiment with HEK293T cells treated with interferon-α for 48 h; 1419 proteins were identified in the experiment with HeLa cells treated with interferon-β for 24 h. Proteins were quantified by the MaxLFQ algorithm integrated in the MaxQuant software. A minimum of two ratio counts from at least one unique peptide was required for quantification. Further data analysis was performed with the Perseus software (version 1.6.2.3) after loading the proteinGroups table from MaxQuant. Hits identified in the reverse database, only identified by modification site and contaminants were removed and protein LFQ intensities were log2 transformed. Replicate samples were grouped, proteins with less than three or four valid values in at least one group were removed and missing values were imputed from a normal distribution around the detection limit to compile a list of quantified proteins. In the virotrap experiment, 613 proteins were quantified. In the GST pull-down experiments, 595 quantified proteins in the experiment with THP-1 cells, 415 with HeLa cells, and 933 with Hek293T cells. In the IP experiments, 223 proteins were quantified in the experiment with HEK293T cells treated with interferon-α for 24 h; 2362 proteins were quantified in the experiment with HEK293T cells treated with interferon-α for 48 h; 547 proteins were quantified in the experiment with HeLa cells treated with interferon-β for 24 h. On the quantified proteins, for each ISG15 bait a t-test was performed for a pairwise comparison with the control condition to reveal specific ISG15 interaction partners. The results of these t-tests are shown in the volcano plot in Fig. 1 and Supplementary Figs. 1-3. For each protein, the log2 (ISG15/control) fold change value is indicated on the X-axis, while the statistical significance (−log p-value) is indicated on the Y-axis (Supplementary Data 1). Proteins outside the curved lines, set by an FDR value of 0.001 and an S0 value of 2 in the Perseus software, represent specific ISG15 interaction partners.

**Generation of RNF213 knockout mice.** B6J-RNF213em1Irc mice were generated using the CRISPR/Cas9 system. Synthetic Alt-R® CRISPR-Cas9 crRNA (Integrated DNA Technologies) with protospacer sequences 5′ CAGAGCTTCGGAACTTTG CT 3′ and 5′ TGTGCCCCTCATCAACCGTC 3′ were duplexed with synthetic Alt-R® CRISPR-Cas9 tracrRNA (Integrated DNA Technologies). cr/tracrRNA duplexes (100 ng/μl) were complexed with Alt-R® S.p. Cas9 Nuclease V3 (500 ng/μl) (Integrated DNA Technologies). The resulting RNP complex was electroporated into C57BL/6 J zygotes using a Nepa21 electroporator with electrode CUY501P1-1.5 using following electroporation parameters: poring pulse = 40 V; length 3.5 ms; interval 50 ms; No. 4; D. rate 10%; polarity+ and transfer pulse = 5 V; length 50 ms; interval 50 ms; No. 5; D. rate 40%; polarity±. Electroporated embryos were incubated overnight in Embryomax KSOM medium (Merck, Millipore) in a CO2 incubator. The following day, two-cell embryos were transferred to pseudopregnant B6CBAF1 foster mothers. The resulting pups were screened by PCR over the target

region using primers 5′ AGTTTCTTGATCTCTTCCCC 3′ and 5′ CTCCTCCGT CAGATCCCTA 3′. PCR bands were Sanger sequenced to identify the exact nature of the deletion. Mouse line B6J-RNF213em1Irc contains an allele with a deletion of 2854 bp (chr11+: 119440493-119443346) in exon ENSMUSE00000645741 resulting in a frameshift and premature stopcodons (Supplementary Fig. 11).

**Statistics and reproducibility.** Statistical tests, significant p values and number of replicates are indicated in the figure legends and are briefly described here. Western blot data are representative of three independent experiments. Proteomics experiments were performed with ≥3 biological replicates. HSV-1 infection experiments were performed with four biological replicates in three independent experiments unless otherwise noted. RSV infection experiments were performed with three biological repeats. CV and *Listeria monocytogenes* infection experiments were performed with three independent repeats, unless otherwise noted. Mouse experiments were performed with ≥4 animals. Non-parametric Mann–Whitney, one-tailed and two-tailed unpaired t-tests or ANOVA tests were performed using GraphPad Prism 9.2.0. P value thresholds used for the statistical tests corresponds to *$p < 0.05$, **$p < 0.01$, ***$p < 0.001$, and ****$p < 0.0001$. The values of single datapoints and the exact p values are indicated in the source data.

**Reporting summary.** Further information on research design is available in the Nature Research Reporting Summary linked to this article.

## Data availability
The raw proteomics data have been deposited in the PRIDE database under accession codes PXD018345 and PXD018346. Data supporting the findings of this manuscript are available within the article, the Supplementary Information and the Source Data files or are available from the corresponding authors upon reasonable request. Source data are provided with this paper.

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

## Acknowledgements

We are grateful to Prof. Akio Koizumi for providing a 3xFLAG-RNF213 expression plasmid, to Prof. Pascale Cossart for sharing strains and mutants of *Listeria monocytogenes* and *Listeria* EF-TU and R11 antisera, to Prof. Stacey Efstathiou for providing a GFP-expressing HSV-1 strain and to Prof. Gary Cohen for providing the HSV-1 VP5 antisera. We thank Evelyn Plets and Annelies Van Hecke for help with the genotyping and in vitro infection experiments. A.B., K.P.K., and F.I are supported by ERANET Infect-ERA BacVIRISG15. The German Research Foundation funded this project with grant KN590/7-1 to K.P.K., BE6335/6-1 to A.B., CRC/TR167, project B16 to K.P.K. and A.B. as well as CRC 1292, project 02 to A.B. A.B. receives support from the Foundation for Experimental Biomedicine Zurich, Switzerland. C.B. is supported by International Max Planck Research School for Infectious Diseases and Immunology (IMPRS-IDI), Berlin. F.I. is supported by an Odysseus type 2 grant from the Research Foundation Flanders (G0F8616N). B.D. is supported by an Odysseus type 1 Grant of the Research Foundation Flanders (3G0H8318) and a starting grant from Ghent University Special Research Fund (01N10319). This work was supported by an NIH grant 1R35GM137961 to L.R. X.S. acknowledges support from the FWO EOS project VIREOS and a BOF-UGent GOA project. N.C. is supported by the European Research Council (ERC) Consolidator Grant GlycoTarget (616966), and S.S. is supported by Research Grants Council, Hong Kong (17113019) and Wellcome Trust (220776/Z/20/Z).

## Author contributions

F.T. interpreted the Virotrap experiments, performed the ISG15 and RNF213 (proteomics) pull-down analyses, compiled all figures, and wrote and edited the paper. L.M. helped F.T. with the ISG15 and RNF213 pull-down analyses, performed the in vitro *Listeria* infection assays, coordinated the in vitro viral infection assays with K.S. and C.B., and wrote and edited the paper. C.A. performed the in vivo *Listeria* infection assays, helped K.S. with the in vivo RSV infection assays, assisted L.M. with the in vitro *Listeria* infection assays, helped L.R., Y.Z., and M.V. with quantification of the imaging data, and wrote and edited the paper. Y.Z. and M.V. performed all imaging experiments. H.R. helped F.T. with the ISG15 and RNF213 pull-down analyses, performed the cellular fractionation assays, and edited the paper. K.S. performed the RSV infection assays and edited the paper. G.D.M. performed the HSV infection assays. C.B. performed the CVB3 infection assays. Q.W.T. and J.Z. performed the lipid droplet isolation and density centrifugation assays in macrophages. K.L. and D.E. helped F.T. with the ISG15 and RNF213 pull-down analyses. D.D.S. performed the Virotrap experiments. K.B. cloned ISG15 and RNF213 constructs. T.H. generated the RNF213 KO mice. N.F. helped with the generation of the KO mice and edited the paper. N.C. supervised N.F. and edited the paper. X.S. supervised K.S., G.D.M, conceptualized the HSV and RSV infection experiments and helped with editing. B.D. supervised C.A and helped with editing the paper and interpretation of the data in relation to MMD. K.P.K. helped with the initial interpretation of the Virotrap experiments and edited the paper. A.B. supervised C.B., conceptualized the CVB3 infection experiments, and helped with editing. S.S. supervised Q.W.T. and J.Z., conceptualized and helped with the lipid droplet isolation and density centrifugation assays as well as editing. L.R. supervised Y.Z. and M.V., conceived and interpreted the imaging experiments, generated figures, and helped with writing and editing of the paper. S.E. supervised D.D.S. and G.D.M., conceived, designed and interpreted the Virotrap experiments, and helped with writing and editing of the paper. F.I. initiated the project, supervised F.T., L.M., and C.A., analyzed the proteomics data, managed data compilation and data interpretation, generated figures, and wrote and edited the paper.

## Competing interests

The authors declare no competing interests.
