## [Peer Review File · Nature Communications]

Ring Finger Protein 213 Assembles into a Sensor for ISGylated Proteins with Antimicrobial ActivityEditorial Note: Parts of this peer review file have been redacted as indicated to maintain the confidentiality of unpublished data.

REVIEWER COMMENTS

Reviewer #1 (Remarks to the Author):

The manuscript submitted by They and colleagues entitled “Ring Finger Protein 213 assembles into a sensor for ISGylated proteins with antimicrobial activity” utilizes VLP trapping technology to identify RNF213 as an ISG15 binding protein. Further characterization revealed that RNF213 oligomerizes and localizes on lipid droplets and that it has antimicrobial activity against *Listeria* and also modest activity against RSV, HSV-1, and CVB3 viruses. There are several striking findings in this manuscript including the identification of RNF213 as a significant binding protein for ISG15 and its ability to bind to ISGylated proteins. In addition, the antimicrobial phenotype shown against *Listeria* is striking including the dramatic increase in bacterial loads in mice in which RNF213 has been knocked out. Overall, this paper makes an important contribution to our understanding of ISG15 biology. There are however a few concerns that should be addressed.

1) While the *Listeria* phenotype is quite convincing the data on antiviral activity of RNF213 is less convincing. The changes in CVB3 are 3-5 fold and the changes in HSV-1 and RSV are even less impressive. One question is whether this is due to the knockdown of RNF213 shown in figure 4C and 4K being only partial. The authors have generated a RNF213 knockout mouse. Are the differences more dramatic in cells derived from the knockout mouse? What about in vivo phenotypes? Since these mice are clearly available this should be evaluated. If the phenotypes are not more striking with the knockout cells or mice, then the paper would be strengthened by focusing on the *Listeria* phenotype.

2) In the *Listeria* model the authors very nicely draw a link between ISG15, RNF213, and its antimicrobial activity in Figure 5C where they demonstrate that overexpression of RNF213 decreases intracellular bacteria, but only in WT cells, not in cells lacking ISG15. If the authors demonstrate a more robust anti-viral phenotype, they should perform a similar experiment in their viral model(s) to draw the same conclusion.

3) In figure 6 the authors demonstrate co-localization of *Listeria* and RNF213 (Fig 6C). Does ISG15 also co-localize in these cells?

4) In figure 8C the authors demonstrate that the %bacteria within a cells is reduced with full length RNF213 but not with the ΔC mutant however the images in figure 8D seem to show little if any infection of the ΔC expressing cells with mCherry-*Listeria*. Are the cells expressing the RNF213 ΔC mutant resistant to infection? Please clarify.

5) In addition, the quality of the microscopy images shown in figure 8D should be improved.

6) Please clarify if the RNF213 ΔC mutant has been shown to lack ubiquitination activity or if this is only predicted? If previously shown this reference should be cited.

Reviewer #2 (Remarks to the Author):

Summary

The IFN-stimulated gene 15 (ISG15) is a ubiquitin-like protein which, by a process known as ISGylation, can covalently modify other proteins and counteract viral infections. Here, They and colleagues, using an elegant system called “virotrap”, identified the RNF213 protein, an AAA+ATPase complex reported as risk factor on Moyamoya disease, as a new ISG15-protein interactor, and describe a novel sensor platform to ISGylated proteins from lipid droplets. Authors claim that upon type I IFN stimulus (or pathogen stimulus), RNF213 protein undergoes oligomerization on lipid droplets in a process dependent of ISGylation to exerts its antimicrobial activity. They found that RNF213-ISG15 sensor limits in vitro infection of herpes simplex virus 1 (HSV-1), human respiratory syncytial virus (RSV) and coxsackievirus B3 (CVB3), as well as controls in vitro and in vivo infection with the intracellular bacteria *Listeria monocytogenes*. Thus, the current study on the RNF213 protein provides information of a novel effector protein on the ISGylation process to counteract microbial invasion, consistent with similar findings which have reported the linking of ISGylation to antimicrobial responses. This is an exciting manuscript and I am generally in support of publication. However, the authors do not provide evidence with respect to the mechanistic process about how this RNF213-ISG15 sensor is limiting the viral as well as bacterial infection. While this is somewhat beyond the scope of the paper, there are some key experiments that should be done here to clarify the observations.

Major comments

1. Figure 5. Results in A and B show that deficiency of RNF213 (as well as ISG15) promotes increased bacterial numbers (in terms of % of bacteria inside cells), and their overexpression has therefore decreased bacterial numbers. According to these observations, authors suggest that RNF213-ISG15 sensor exerts an antimicrobial function during Lm infection, limiting the bacterial numbers inside cells. However, these observations might have an alternative explanation: the phagocytosis process could be affected by the presence/absence of these two molecules, directly affecting the number of bacteria inside the cells. The authors should consider performing a phagocytosis assay to discard this alternative possibility.

2. Figure 6. Results in A show colocalization of RNF213 molecule on lipid droplets during Lm infection, and although there were not differences between control and infected conditions, two concerns come up here. First one, under the title “IFN-I induces RNF213 ISGylation and oligomerization on lipid droplets” (line 221), authors conclude that localization and oligomerization

of RNF213 molecule on lipid droplets (as well as ISGylation) is a process induced by type I IFN. Keeping this in mind, how can it be explained that in Figure 6A, localization of RNF213 molecule occurs on lipid droplets in the untreated condition?

3. Figure 6A, panel showing “neutral lipid droplet” in Lm condition. What is observed is a decreased number of lipid droplets (puncta) with respect to the untreated control. Is this an inaccurate selection of representative images, or could it be a consequence of the bacterial infection? It is reported that several intracellular pathogens hijack host cell nutrient reserves (such lipids) to promote their own survival. It would be interesting test if Lm is indirectly targeting itself with the RNF213-ISG15 sensor trying to consume lipids from lipid droplets.

4. Line 297, “on average approximately 40% of intracellular Listeria co-localized with RNF213 (Fig6C-D)”. What is this subset of intracellular bacteria? This is an important question since Listeria can grow within the cytosol and in vacuoles during infection. For a subset of bacteria to be targeted by RNF213 one presumes this is affecting their fate. The authors should examine other markers to determine if the RNF213+ bacteria are in vacuoles (LAMP1+), autophagosomes (LC3+) or in the cytosol (F-actin positive). Correlative light-electron microscopy should also be considered, if possible. Are the motile bacteria (on comet tails), which are associated with Listeria growth and spread, positive for RNF213? Or is it a non-replicating population trapped in vacuoles/phagosomes?

5. ISG15 can modulate the type I interferon pathway, which impacts both viral and Listeria infection. Does RNF213 impact type I interferon during infections here? At a minimum, the authors should blot samples to examine interferon stimulated gene expression.

Minor comments

1. Line 290, I’m not sure “profoundly” is accurate, the word can be removed

2. In line 306, authors claim a central role of RNF213 in immune cell function and intracellular bacterial clearance, given RNF213-/- mice showed a protective effect against Lm infection respect to WT mice, and an increased % of bacteria inside cells in vitro, respectively. However, in the current study, there is no immunological evidence of how RNF213 activity is governing immune response in vivo, like cytokines produced, macrophage and/or neutrophil infiltration or if there is a specific effector lymphocytic response, such as CD4 or CD8. Neither in vitro, such which mechanism is

involved to directly limits viral/bacterial pathogen. If authors are agreed, this conclusion must be adapted according to the evidence obtained from current experiments.

3. Line 422 "Co-localization of proteins with intracellular bacteria is described as an antibacterial strategy known as xenophagy". This is not an accurate definition of xenophagy . It is also worth noting that cytosolic proteins can kill bacteria without autophagy.

4. Line 429 "Interestingly, wild-type bacteria evade xenophagy through cell-to-cell spread 83,". This is not accurate: Listeria evades xenophagy through many mechanisms, involving actA, plcA, plcB, etc. The paper (ref 83) refers to a role for ActA and its role in actin polymerization, not in cell-to-cell spread per se.

Reviewer #3 (Remarks to the Author):

In the manuscript by Francis Impens and colleagues, the authors describe a novel sensor for ISG15ylated proteins in the context of cellular antimicrobial activity. They characterise the RING finger protein RNF213 as a novel ISG15 binding protein that accumulates at sites of intracellular pathogens.

The authors have used a clever screening approach based on mass spectrometry & proteomics in combination with a viral trap ("virotrap") method to identify the RING finger ubiquitin E3 ligase as one of the major non-covalent ISG15 binding constituents.

The authors subsequently provide evidence for the role of RNF213 in cellular antimicrobial defence mechanisms. This includes co-localisation studies of RNF213 on sites of infection as well as functional evidence of its direct functional involvement.

Specific points:

1. This study provides novel insights into the cellular interferone stimulated gene (ISG) response, in this case against listeria infection. RNF213 has been identified to be a player in this process. This is novel and interesting new information that definitively should be considered to be reported at the level of this journal.

2. The discovery of RNF213 as a major interactor was performed by an expert group in proteomics, using a mass spectrometry based approach. The experimental method well described and the essential details have been provided.

In particular, the authors have made use of the "Virotrap" method using a fused ISG15-Gag(HIV) construct, which captures ISG15 binding proteins and packs them up into vesicles that get secreted, thereby simplifying the isolation process. With the proper controls (which the authors provided), this reflects an elegant application for the isolation of specific interactors, given that there are no space/packing issues as space and steric hindrance in such vesicles must be limiting.

Could the authors comment on this potential limitations?

3. The authors have followed protocol and deposited their mass spectrometry data into a public repository (PRIDE). However, it is somewhat unclear why they did this in two different deposits, what is what? - please clarify.

4. As the authors stated, RNF213 is a ubiquitin E3 ligase of the RING finger family. At this stage, it is not clear what its role is as a ubiquitin E3 ligase in this process. Is there a possible interplay and/or regulation by ISG15 binding at these critical sites described by the authors?

Note: changes in the main text related to the reviewer comments are indicated in **green**.

Reviewer #1

The manuscript submitted by They and colleagues entitled “Ring Finger Protein 213 assembles into a sensor for ISGylated proteins with antimicrobial activity” utilizes VLP trapping technology to identify RNF213 as an ISG15 binding protein. Further characterization revealed that RNF213 oligomerizes and localizes on lipid droplets and that it has antimicrobial activity against *Listeria* and also modest activity against RSV, HSV-1, and CVB3 viruses. There are several striking findings in this manuscript including the identification of RNF213 as a significant binding protein for ISG15 and its ability to bind to ISGylated proteins. In addition, the antimicrobial phenotype shown against *Listeria* is striking including the dramatic increase in bacterial loads in mice in which RNF213 has been knocked out. Overall, this paper makes an important contribution to our understanding of ISG15 biology. There are however a few concerns that should be addressed.

1) While the *Listeria* phenotype is quite convincing the data on antiviral activity of RNF213 is less convincing. The changes in CVB3 are 3-5 fold and the changes in HSV-1 and RSV are even less impressive. One question is whether this is due to the knockdown of RNF213 shown in figure 4C and 4K being only partial. The authors have generated a RNF213 knockout mouse. Are the differences more dramatic in cells derived from the knockout mouse? What about *in vivo* phenotypes? Since these mice are clearly available this should be evaluated. If the phenotypes are not more striking with the knockout cells or mice, then the paper would be strengthened by focusing on the *Listeria* phenotype.

Answer: We agree with the reviewer that the antiviral effect of RNF213 is less pronounced compared to the antibacterial effect. Nevertheless, we find this effect to be consistent and significant over three different viral pathogens. Initially, in combination with knockdown of RNF213 we observed increased infection with HSV-1, RSV and CVB3, while overexpression of RNF213 reduced infection with HSV-1. In the meantime, we could also demonstrate that overexpression of RNF213 reduces infection with CVB3 and we have now included these novel data in the revised manuscript (**Supplementary Fig. 7K-M**), further strengthening the antiviral phenotype. Although knockdown in **Fig. 4C** (HSV-1) and **Fig. 4K** (CVB3) was indeed only partial, we have now repeated the experiment with HSV-1 with near complete knockdown (see next point), which confirmed the modest increase in infection as in **Fig. 4A-C** (**Supplementary Fig. 7C-E**).

To evaluate *in vivo* phenotypes as the reviewer requests, we infected our RNF213 KO mice with a mouse-adapted strain of RSV. We observed nearly a twofold increase in viral titers in the lungs of RNF213 KO mice compared to WT littermates, however, this increase did not reach statistical significance. The KO animals lost significantly more body weight and all of these mice had reached the ethical endpoint by day 7 after RSV infection whereas most of the WT littermates survived the challenge. These *in vivo* data is now included in the revised manuscript (**Supplementary Fig. 9F-H**). Thus, we observe a subtle, but clear antiviral phenotype both *in vitro* and *in vivo*.

Although we see why the reviewer would focus on the *Listeria* data, we believe it is highly relevant to keep the viral infection data in the manuscript, especially since it was recently reported that RNF213 contributes to Rift Valley fever (RVF) virus resistance in mice (Houzelstein et al., Mamm Genome 2021). In this study, RNF213 was found by quantitative trait locus (QTL) analysis to be present in a genomic region that defines susceptibility of mice to RVF virus. Without providing further mechanistic insight, the authors show that RNF213-deficient mice show significantly reduced survival times after RVF infection. As a second study that

reports an antiviral phenotype, we now cite this RVF paper both in the results and discussion. We do, however, clearly state that in our experiments the observed antiviral phenotype is less pronounced compared to the antibacterial phenotype and that the underlying mechanisms might be different and ISG15-independent (see next point). We thus made the differences with the antibacterial phenotype more clear in the revised version of the manuscript and hope that the reviewer understands our rationale to keep the viral data in the paper.

2) In the listeria model the authors very nicely draw a link between ISG15, RNF213, and its antimicrobial activity in Figure 5C where they demonstrate that overexpression of RNF213 decreases intracellular bacteria, but only in WT cells, not in cells lacking ISG15. If the authors demonstrate a more robust anti-viral phenotype, they should perform a similar experiment in their viral model(s) to draw the same conclusion.

Answer: We performed the suggested experiment in the HSV-1 infection model. Since with HSV-1 infection, the phenotype is more pronounced with knockdown of RNF213 instead of overexpression (**Fig. 4A-E**), we combined knockdown of RNF213 with HSV-1 infection in either WT or ISG15 KO HeLa cells to address this question. RNF213 knockdown significantly reduced infection both in WT and ISG15 KO cells, indicating that the antiviral activity of RNF213 is not dependent on ISG15. These data are now included in **Supplementary Fig. 7C-H**. Together with the more subtle antiviral effect, these results indicate that the underlying mechanisms by which RNF213 counteracts bacterial and viral pathogens are probably different, something we now clearly state in the discussion.

3) In figure 6 the authors demonstrate co-localization of listeria and RNF213 (Fig 6C). Does ISG15 also co-localize in these cells?

Answer: We initially co-stained for ISG15 in these experiments, but could only observe rare instances of co-localization with RNF213 and lipid droplets. We believe that immunofluorescence imaging with the current commercially available antibodies is not sensitive enough to differentiate between ISGylated proteins bound to RNF213 (or ISGylated RNF213 itself) versus free cytosolic ISG15. We thus were forced to rely on biochemical purification of lipid droplets as the more sensitive method to differentiate between ISG15 conjugates versus cytosolic ISG15 (**Fig. 2** and **Fig. 3**). Furthermore, HeLa cells (used for IF and infection experiments) may not have the same ratio of ISGylated RNF213 as THP-1 cells. Since macrophages have increased microbicidal effects, ISGylation of targets could be distinct in this cell type. We are interested in following up on the cell type specific properties in future work. Finally, for our infection experiments in HeLa cells we did not pretreat the cells with oleic acid and this could have led to increased ISG15 in the cytosol versus the lipid droplet. Ultimately, these technical issues prevented us from imaging ISG15 in other experiments.

4) In figure 8C the authors demonstrate that the % bacteria within a cells is reduced with full length RNF213 but not with the ΔC mutant however the images in figure 8D seem to show little if any infection of the ΔC expressing cells with mCherry-listeria. Are the cells expressing the RNF213 ΔC mutant resistant to infection? Please clarify.

Answer: To address this point we repeated the invasion assay shown in **Fig. 8C**, but on an early time point (1 hour post infection) to assess bacterial uptake. This experiment showed no difference in bacterial entry between cells expressing WT RNF213 and RNF213 ΔC . These data is now shown in **Supplementary Fig. 8C-D**.

We agree with the reviewer that the representative image shown in **Fig. 8D** suggests that the transfected (green) cell contains less bacteria compared to the neighbouring untransfected cell. We tried to quantify the amount of bacteria in the transfected vs. untransfected cells in this dataset, but the high transfection rates did not leave enough untransfected cells for reliable quantification. It is well-known that plasmid transfection increases cellular resistance to *Listeria* by sensing of cytosolic DNA and downstream upregulation of ISGs through cGAS and STING. Hence, it was difficult to select a representative image not showing this effect and therefore we addressed this comment by the early time point invasion assay. However, we also suspect that the image quality was reduced in the compressed PDF and have improved the current figure quality as requested so that the bacteria in the cell are easier to visualize. Finally, while our paper was in review, the group of Felix Randow reported RNF213 as an E3 ligase that ubiquitylates cytosolic *Salmonella*, directly on LPS (Otten et al, Nature 2021). Similar to our *Listeria* data, RNF213 was shown to restrict *Salmonella* proliferation in MEF cells, also without any effect on bacterial entry. We now cite this paper in the results and discussion. Of note, a recent commentary on the Randow paper (Damgaard and Pruneda, Mol Cell 2021) also cited our manuscript from bioRxiv. Together both studies show the key role of RNF213 in restricting Gram- as well as Gram+ intracellular bacteria.

5) In addition, the quality of the microscopy images shown in figure 8D should be improved.

Answer: The reviewer is likely referring to the quality of the figure in the compiled PDF file. We also noticed that the resolution of some images was reduced in this file, but could not control this in the online editorial system. We also uploaded all figures separately and in high resolution in the editorial system and hope that the quality of our figures can be assessed through download of the individual figure.

6) Please clarify if the RNF213 Δ C mutant has been shown to lack ubiquitination activity or if this is only predicted? If previously shown this reference should be cited.

Answer: The lack of ubiquitination activity of our RNF213 Δ C mutant is only predicted based on the RNF213 cryo-EM structure (Ahel et al. eLife 2020). However, the recent paper from the Randow lab shows that ubiquitylation activity of RNF213 relies on an RZ finger (RNF213-ZNFX1 finger) that is present in the E3 shell (one part of the E3 module, Otten et al, Nature 2021), something that is mechanistically supported by unpublished work (Ahel et al, bioRxiv, doi: <https://doi.org/10.1101/2021.05.10.443411>). Based on these results, it can be assumed that complete deletion of the E3 module will lack any ubiquitination activity. We have adjusted the language in the results to make this more clear and included citations to the aforementioned papers.

Reviewer #2

Summary

The IFN-stimulated gene 15 (ISG15) is a ubiquitin-like protein which, by a process known as ISGylation, can covalently modify other proteins and counteract viral infections. Here, They and colleagues, using an elegant system called "virotrap", identified the RNF213 protein, an AAA+ATPase complex reported as risk factor on Moyamoya disease, as a new ISG15-protein interactor, and describe a novel sensor platform to ISGylated proteins from lipid droplets. Authors claim that upon type I IFN stimulus (or pathogen stimulus), RNF213 protein undergoes oligomerization on lipid droplets in a process dependent of ISGylation to exerts its antimicrobial activity. They found that RNF213-ISG15 sensor limits in vitro infection of herpes simplex virus 1 (HSV-1), human respiratory syncytial virus (RSV) and coxsackievirus B3 (CVB3), as well as controls in vitro and in vivo infection with the intracellular bacteria *Listeria monocytogenes*. Thus, the current study on the RNF213 protein provides information of a novel effector protein on the ISGylation process to counteract microbial invasion, consistent with similar findings which have reported the linking of ISGylation to antimicrobial responses. This is an exciting manuscript and I am generally in support of publication. However, the authors do not provide evidence with respect to the mechanistic process about how this RNF213-ISG15 sensor is limiting the viral as well as bacterial infection. While this is somewhat beyond the scope of the paper, there are some key experiments that should be done here to clarify the observations.

Major comments

1. Figure 5. Results in A and B show that deficiency of RNF213 (as well as ISG15) promotes increased bacterial numbers (in terms of % of bacteria inside cells), and their overexpression has therefore decreased bacterial numbers. According to these observations, authors suggest that RNF213-ISG15 sensor exerts an antimicrobial function during Lm infection, limiting the bacterial numbers inside cells. However, these observations might have an alternative explanation: the phagocytosis process could be affected by the presence/absence of these two molecules, directly affecting the number of bacteria inside the cells. The authors should consider performing a phagocytosis assay to discard this alternative possibility.

Answer: We addressed this question by an early time-point gentamycin assay to assess entry and uptake, counting intracellular bacteria 1 hour post infection as the result of cellular phagocytosis (and not intracellular replication). The results of this assay are now included in **Supplementary Fig. 8C-D**, showing no significant difference in cellular uptake upon knockdown or overexpression of RNF213. Moreover, while our paper was in review, the group of Felix Randow reported RNF213 as an E3 ligase that ubiquitylates cytosolic *Salmonella*, directly on LPS (Otten et al, Nature 2021). Similar to our *Listeria* data in HeLa cells, reduced levels of RNF213 increased proliferation of *Salmonella* in MEF cells, also without any effect on bacterial entry. We now cite this paper in the results and discussion. Of note, our manuscript on bioRxiv was cited in a recent commentary on the Randow paper (Damgaard and Pruneda, Mol Cell 2021), highlighting that together both our studies show the key role of RNF213 in restricting both Gram- and Gram+ intracellular bacteria.

2. Figure 6. Results in A show colocalization of RNF213 molecule on lipid droplets during Lm infection, and although there were not differences between control and infected conditions, two concerns come up here. First one, under the title “IFN-I induces RNF213 ISGylation and oligomerization on lipid droplets” (line 221), authors conclude that localization and oligomerization of RNF213 molecule on lipid droplets (as well as ISGylation) is a process induced by type I IFN. Keeping this in mind, how can it be explained that in Figure 6A, localization of RNF213 molecule occurs on lipid droplets in the untreated condition?

Answer: This is a very good point. We believe that this discrepancy might be explained by overexpression of GFP-RNF213 (in HeLa cells) in **Fig. 6** versus monitoring of endogenous RNF213 (in THP-1 cells) in **Fig. 2** and **Fig. 3**. Indeed, overexpression of RNF213 might disturb the equilibrium of monomeric (cytosolic) versus oligomeric (lipid droplet) RNF213, overruling endogenous regulation of RNF213 by IFN-I or other stimuli.

This is an important point that we would like to address in the future and something that was largely ignored in previous studies on RNF213, including the work of Sugihara et al., JCB 2019 and Otten et al., Nature 2021. In addition to overexpression of RNF213, also the common use of oleic acid to induce lipid droplet (LD) formation as well as the investigated cell type are parameters that could affect RNF213 localization to LDs (and likely also bacteria). The discrepancy between the effect of RNF213 on LD stabilization reported by Sugihara 2019 and our data presented in **Supplementary Fig. 5** illustrates this point. Sugihara 2019 showed that in the presence of oleic acid the number of LDs in RNF213 KO HeLa cells is drastically reduced compared to WT cells. However, under these conditions we did not observe any difference in the number of LDs in BMDMs from WT and RNF213 KO mice. IFN-I treatment resulted in smaller LDs, but without difference between RNF213 WT and KO cells, and not affecting the number of LDs. Together, these data show that depletion of RNF213 does not lead to reduced stability of LDs in macrophages, in contrast to HeLa cells.

In follow-up work, we aim to investigate the oligomerization and localization of RNF213 in LD and bacterial targeting, using more advanced cellular tools such as chromosomally GFP-tagged RNF213 variants, to properly monitor the dynamic localization of endogenous RNF213. However, such experiments are outside the scope of the present paper.

3. Figure 6A, panel showing “neutral lipid droplet” in Lm condition. What is observed is a decreased number of lipid droplets (puncta) with respect to the untreated control. Is this an inaccurate selection of representative images, or could it be a consequence of the bacterial infection? It is reported that several intracellular pathogens hijack host cell nutrient reserves (such lipids) to promote their own survival. It would be interesting test if Lm is indirectly targeting itself with the RNF213-ISG15 sensor trying to consume lipids from lipid droplets.

Answer: The representative images in **Fig. 6A** were selected to show co-localization of RNF213 with *Listeria*. As a consequence, the images are unfortunately not representative for the number of lipid droplets suggesting that *Listeria* infection leads to a decrease in lipid droplets, while in fact the opposite is true. Indeed, we have now quantified the number lipid droplets in this dataset and found that *Listeria* infection significantly increased the number of lipid droplets/cell, in line with previous results on *E. coli* or *Salmonella* infected cells (Bosch & Sanchez-Alvarez et al., Science 2020). These data are now shown in **Supplementary Fig. 8F** and a sentence has been added to the corresponding results section.

4. Line 297, “on average approximately 40% of intracellular *Listeria* co-localized with RNF213 (Fig6C-D)”. What is this subset of intracellular bacteria? This is an important question since *Listeria* can grow within the cytosol and in vacuoles during infection. For a subset of bacteria to be targeted by RNF213 one presumes this is affecting their fate. The authors should examine other markers to determine if the RNF213+ bacteria are in vacuoles (LAMP1+), autophagosomes (LC3+) or in the cytosol (F-actin positive). Correlative light-electron microscopy should also be considered, if possible. Are the motile bacteria (on comet tails), which are associated with *Listeria* growth and spread, positive for RNF213? Or is it a non-replicating population trapped in vacuoles/phagosomes?

Answer: We thank the reviewer for these excellent experimental suggestions. We decided to first take advantage of existing bacterial mutants to address the underlying question of whether the bacteria targeted by RNF213 are within membranes or have escaped into the cytosol. We used a triple mutant which lacks phospholipase A, phospholipase B, and hly which encodes the major *Listeria* hemolysin, Listeriolysin O. This triple mutant remains within the vacuole and thus if RNF213 is recruited to the surface it would indicate that RNF213 is docking on a membrane surrounding the bacteria. We instead observed that RNF213 never stained this triple mutant and solely localized to the surface of cytosolic wild-type bacteria. There were some instances of RNF213 positive vesicles in the vicinity of the triple mutant, however we never observed co-localization with the bacterial pole or surface as observed with the wild-type strain. Co-staining for F-actin as suggested by the reviewer showed some instances of WT bacteria with the septum between two dividing bacteria covered with RNF213 (which is known to lack ActA), but we did not observe direct co-localization between RNF213 and actin. Thus, it appears that cytosolic bacteria which can be replicating are the RNF213 target, the interplay of this targeting with bacterial motility will be of interest to dissect in future work. These data are now included along with the other microscopy data in **Fig. 6E**.

We subsequently stained for RNF213 co-localization with LC3 as also suggested by the reviewer. While revising our manuscript Felix Randow's group published that RNF213 acts as a ubiquitin E3 ligase to modify the LPS of cytosolic *Salmonella*. *Listeria monocytogenes* is a Gram+ pathogen, and thus does not have LPS, but our complementary study suggests that RNF213 is a broad antibacterial E3 ligase. Ubiquitin modification of the bacterial surface or membrane remnants are sensed by linker proteins like p62 or NDP52 and bring the cargo into contact with LC3. Unfortunately, we were unable to properly visualize endogenous LC3 due to issues with background staining and incompatible fixation methods. For our autophagy work, we routinely use GFP-LC3 but since RNF213 is also labeled with GFP we instead stained with the ubiquitin-FK2 antibody which marks poly and mono ubiquitin chains. FK2 covered the surface of the bacteria adjacent to RNF213 staining, suggesting that RNF213 is able to ubiquitinate as yet unknown substrates on the surface of Gram-positive bacteria. These data are now shown in **Supplementary Fig. 8G**. Unlike *Salmonella*, the majority of cytosolic *Listeria* evades targeting by autophagy using actin-based motility. It appears that overexpression of RNF213 prior to infection tips the balance in favor of xenophagic clearance of bacteria, even for wild-type bacteria that can evade autophagy through ActA and phospholipase A, among other mechanisms. This could also account for the extreme susceptibility of RNF213 deficient animals to *Listeria monocytogenes* infection.

Finally, as the reviewer suggested we co-stained with LAMP1 to see whether RNF213 co-localized with membrane remnants or a membrane on bacteria. As would be expected based on the published literature the co-localization of LAMP1 with RNF213 and bacteria was fairly rare at six hours post infection as the

majority of wild-type bacteria are not targeted by LC₃ (work from many groups suggests that the proportion would be higher with the ActA mutant). Our data show some areas of co-localization of RNF213 and LAMP1 in the vicinity of bacteria, however the signal from the bacteria in this case is fairly dim, potentially indicating that this particular bacterium is targeted by autophagosomes or lysosomes. In the future we wish to carefully dissect the fate of RNF213 targeted bacteria with regards to the substrate of RNF213 on *Listeria* and the adaptor molecules that link ubiquitin to autophagy; however, given the depth and breadth of the current manuscript we believe these experiments merit their own future study and are beyond the scope of this current work. Thus, we have included LAMP1 recruitment as requested by the reviewer in **Reviewer Fig. 1** below, but would like to reserve these data for future work and not include them in the revised manuscript.

[Redacted]

5. ISG15 can modulate the type I interferon pathway, which impacts both viral and Listeria infection. Does RNF213 impact type I interferon during infections here? At a minimum, the authors should blot samples to examine interferon stimulated gene expression.

Answer: Human ISG15 can indeed modulate IFN signaling through stabilization of USP18, which dampens JAK-STAT signaling and the expression of ISGs. In some of our infection experiments we have monitored ISG expression (e.g. expression of MXA during HSV-1 infection as shown in **Fig. 4C**), but we never observed any effects of RNF213 knockdown or overexpression on ISGs. To evaluate potential effects in a systematic way, as suggested by the reviewer we have now performed immunoblots against several ISGs (including ISG15, STAT1, P-STAT1, IFIT-1, Mx1 and OAS3) in HeLa cells treated with interferon- β in combination with overexpression or knockdown of RNF213. These results are now included in **Supplementary Fig. 7A-B** showing no difference in expression of these ISGs with changing RNF213 expression levels. Moreover, we also analyzed these cells by shotgun proteomics, again showing efficient induction of ISGs in all conditions. We intend to include these proteomics data in a follow-up study, but are happy to share it with the reviewer in **Reviewer Fig. 2** below.

[Redacted]

Minor comments

1. Line 290, I'm not sure "profoundly" is accurate, the word can be removed

Answer: We agree with the reviewer and have adjusted the text accordingly.

2. In line 306, authors claim a central role of RNF213 in immune cell function and intracellular bacterial clearance, given RNF213^{-/-} mice showed a protective effect against Lm infection respect to WT mice, and an increased % of bacteria inside cells in vitro, respectively. However, in the current study, there is no immunological evidence of how RNF213 activity is governing immune response in vivo, like cytokines produced, macrophage and/or neutrophil infiltration or if there is a specific effector lymphocytic response, such as CD4 or CD8. Neither in vitro, such which mechanism is involved to directly limits viral/bacterial pathogen. If authors are agreed, this conclusion must be adapted according to the evidence obtained from current experiments.

Answer: A detailed characterization of the (aberrant) immune response of RNF213 KO mice against *Listeria* is indeed something that is currently ongoing and outside the scope of the present study. Hence, we changed this sentence from "*This profound difference in bacterial load, particularly in the spleen, reveals the central importance of RNF213 in immune cell function and intracellular bacterial clearance.*" to "*This profound difference in bacterial load, particularly in the spleen, reveals the central role of RNF213 in the host defense against bacterial infection.*"

3. Line 422 "Co-localization of proteins with intracellular bacteria is described as an antibacterial strategy

known as xenophagy". This is not an accurate definition of xenophagy. It is also worth noting that cytosolic proteins can kill bacteria without autophagy.

Answer: We thank the reviewer for pointing out this inaccuracy. We have now changed this sentence to "*Proteins that co-localize with intracellular bacteria are often involved in an antibacterial strategy known as xenophagy*".

4. Line 429 "Interestingly, wild-type bacteria evade xenophagy through cell-to-cell spread 83,". This is not accurate: *Listeria* evades xenophagy through many mechanisms, involving actA, plcA, plcB, etc. The paper (ref 83) refers to a role for ActA and its role in actin polymerization, not in cell-to-cell spread per se.

Answer: We agree with the reviewer. This paragraph is rewritten in light of the recent paper from the Randow lab and the concerned sentence now reads as follows: "*Future studies should investigate the fate of RNF213-decorated Listeria in more depth, especially since in contrast to Salmonella, cytosolic Listeria can evade xenophagy through various mechanisms involving major virulence factors such as LLO, PlcA and ActA (67-70)*".

Reviewer #3

In the manuscript by Francis Impens and colleagues, the authors describe a novel sensor for ISG15ylated proteins in the context of cellular antimicrobial activity. They characterise the RING finger protein RNF213 as a novel ISG15 binding protein that accumulates at sites of intracellular pathogens. The authors have used a clever screening approach based on mass spectrometry & proteomics in combination with a viral trap ("virotrap") method to identify the RING finger ubiquitin E3 ligase as one of the major non-covalent ISG15 binding constituents. The authors subsequently provide evidence for the role of RNF213 in cellular antimicrobial defence mechanisms. This includes co-localisation studies of RNF213 on sites of infection as well as functional evidence of its direct functional involvement.

Specific points:

1. This study provides novel insights into the cellular interferone stimulated gene (ISG) response, in this case against listeria infection. RNF213 has been identified to be a player in this process. This is novel and interesting new information that definitively should be considered to be reported at the level of this journal.

Answer: We are very grateful for the reviewer's support and enthusiasm about our findings.

2. The discovery of RNF213 as a major interactor was performed by an expert group in proteomics, using a mass spectrometry based approach. The experimental method well described and the essential details have been provided. In particular, the authors have made use of the "Virotrap" method using a fused ISG15-Gag(HIV) construct, which captures ISG15 binding proteins and packs them up into vesicles that get secreted, thereby simplifying the isolation process. With the proper controls (which the authors provided), this reflects an elegant application for the isolation of specific interactors, given that there are no space/packing issues as space and steric hindrance in such vesicles must be limiting. Could the authors comment on this potential limitations?

Answer: The diameter of Virotrap particles is around 100-120 nm (**Reviewer Fig. 3A** for an EM image of Virotrap particles) which implies that the cavity is around 70 nm (because of GAG and bait). This still accommodates sizable complexes (the large 26S proteasome is 45 nm vs x 20 nm while human ribosomes are ~28 nm). In fact, the entrapped RNF213 protein is the ~20th largest protein in the human proteome.

Reviewer Fig. 3B shows the RNF213 peptides (in red bold) detected in the Virotrap experiment confirming that the full size isoform of the protein was trapped in the particles. While a flexible linker is included in the bait construct, sterical hindrance cannot be fully excluded although it can be expected that out of ~3000 (ISG-15) bait copies in a particle some variation in the exposed sides can be expected. The N-terminal part which is coupled to the GAG protein will likely not be fully tested in this configuration of Virotrap.

(A)

Reviewer Figure 3. (A) EM image of Virotrap particles. Scale bar: 100 nm. (B) Amino acid sequence of the 591 kDa RNF213 protein (isoform 1) with identified peptides in the Virotrap experiment mapped on the sequence.

(B)

```

mecpsqhvsketpkfscqgerlppaapiadsennntmasasagemecggelkeeggcpfpGSDSWQENPEEPCSK
ASWTVOESKSKKRRKKkknksasselalplspaspchl1l1snpwgqdtalphsqagqsptgqpsppqgattpleg
dqlsaptevgsplqaqalgeagvatgseagsppqfdhTEGEDQDASIFSGGRLSQEGTGPPTSGAGHRSRTEDAAGE
LLLEPESKSGSsepptelqtteqagagasamavdavaepanavkgagkemkettqrmkqppaTTPFFKTHCQEAETKTKDE
MAAAEEKVGKNEQGEPEDLKKPEGNRSAAAVKNEKQKQEAQVQEKASTLSpgggtvfvfhaiisLHFFPNPDLKHV
FIRGEEFEGESKWSN1Celhytrdldghdrv1veg1vciskhldkypkyviynGESFEYET1YKHQKGEYVNRCL
FIKSSLLgsgdwhqYD1VYMKPHGR1KQVMNH1TDGFRDLVKGqiaaalmldstfslgtwtidnlnsfftqefgc
fldwlcHLLTSDASSPEFHRDL1SHLGI1PQSRLY1vnlcqrmdtrtvtlwalpvl1HCCMELAPRHKDARWQPEDTWA
ALEGLSFPFPREQM1DTSLLQFMREKHLLS1DEPLFERSWF1lplshlmyemfeniehlq1FFAHL1DCLSGIYTR1P
GLEQV1NTQDVQ1vqnvqnilml1rlldtrdykipeealspsyltvclkhieacsstkl1kfyLEL1SAE1VYCRM1R
LLSLVDSAGQDE1TGNNSV1TFFQGT1laatrkrw1revftknml1tssGASFTYVKE1EVWRRLVEIQ1FABHGKESLE1GD
MEWRL1KE1PLSQ1TAYNC1WCD1TGLEDSVAKT1FEKCI1eavssaacsQ1TSL1QGF5YSD1LRKGF1VL1SAV1T1KSWPT
ADNFND1LKHL1TLADVKHFR1Lc1TDEK1LANV1TEDAKR1LA1VADSV1LTKV1GV1D11s1GT1L1LVQLE1L1IKHKNQF1D1W
Q1LREK1SLPQDE1QCAVE1EALDWR1RE1LL1K1K1EKRCV1D1SL1KMG1NVK1H1I1QVDF1GLV1AVR1HS1qd1sskrlnd1T1V1RLS
T1SSNR1QATHY1L1SSQV1QEMAG1K1D1LL1RD1SH1F1QL1FWR1EAE1PL1SEPK1EQE1AAE1L1SEPEE1SERH1LE1LEEV1Y1Q
PSYRF1KH1QD1L1KSGE1V1TAE1D1V1FKD1FV1NKY1TD1L1D1SEL1K1M1CTV1Dh1QD1QR1D1K1D1R1VE1Q1E1R1KH1H1Q1AV1HA1K1V1
lqvkes1g1ngd1fsv1NTL1N1TND1FDD1FR1RE1T1D1Q1N1Q1E1L1QAK1L1Q1D1SEARC1K1GL1QAL1SEK1E1FT1C1W1RE1AL1G1NE
1K1V1FD1LAS1S1AGEND1I1dvdr1vaCF1DAV1Q1YAS1LL1FK1LD1PS1VD1F1AF1MK1H1K1L1K1W1AL1DK1Q1Y1FK1RL1D1CS1AR1N1LE1K1
TVNE1SH1GS1VER1S1SL1L1ATA1N1QR1GI1Y1V1Q1AP1K1G1K1I1S1PD1TV1L1L1L1PE1SP1GS1HEE1S1RE1YS1LEE1V1K1EL1N1K1M1ms1g1Kd
rnn1TEVER1SE1V1FC1SVQR1LSQA1FD1L1SAG1N1L1r1rtw1am1ay1c1sp1k1g1v1s1q1md1f1gd1l1v1te1KEGG1V1TE1LL1A1L1C1R1Q1M
EH1D1L1SK1R1F1V1TK1rme1fy1n1fy1taeq1lv1st1el1r1k1p1p1s1daa1l1tm1s1f1ks1n1ct1r1d1v1r1ASV1GG1SEAA1RY1M1R1V1
MEEL1P1ML1SE1F1S1V1D1K1R1L1ME1Q1SMR1CL1PA1FL1P1D1cd1d1et1l1gh1cl1ah1ag1m1g1s1p1ver1cl1pr1gl1v1q1g1n1l1v1c1g1ns1v
1paal1av1ym1q1ps1p1t1y1de1v1l1ct1p1att1fe1e1val1l1rr1cl1l1g1h1k1y1s1l1fad1q1s1e1var1q1ae1l1f1m1l1ct1q1h
redy1gl1vm1vc1gd1W1EH1CY1L1S1A1S1F1Q1K1V1T1P1Q1M1PL1EQ1Y1LAG1H1V1F1K1GT1S1AA1V1F1ND1R1C1G1v1i1va1e1er1ag1v1k1s1l
y1kr1l1h1k1k1M1QL1N1K1V1L1K1R1L1R1D1P1Q1D1E1S1V1L1G1ALL1P1LD1AQ1G1K1v1p1l1f1h1d1V1TS1V1S1Q1V1G1W1F1L1K1L1L1Q1V1M
D1NG1K1M11rnp1ch1y1e1iler1RT1S1V1PS1RS1S1AL1R1TR1V1Q1F1S1FD1L1F1FK1V1CR1P1KE1V1D1M1EL1S1AL1R1SD1E1Pg1md1w1ef1c
ser1f1qr1y1p1y1r1f1n1q1d1d1t1f1y1q1e1s1w1e1t1p1ee1cl1q1h1f1h1c1V1N1P1S1W1EL1N1F1AR1F1L1D1R1CES1AL1C1N1p1afi
gd1lrg1fk1k1fv1t1f1m1f1mar1F1AT1P1S1L1T1S1D1Q1S1PK1H1V1T1MD1G1V1RED1LAP1S1lr1kr1w1es1eph1y1f1f1nd1td1tm1f1f1g
h1l1p1n1g1s1v1d1L1SH1L1G1K1V1K1R1D1V1M1R1D1Y1Q1L1L1Q1RV1F1V1D1FK1L1P1RH1K11er1L1CL1FL1G1P1Q1AT1D1M1Y1R1E1R1V1A1F1AN1K1O1H1Q1L
L1K1L1A1E1M1F1R1C1G1P1V11m1get1G1C1K1R1L1K1F1L1S1DL1R1G1T1NAD1T1K1L1K1V1H1G1T1T1AD1M1Y1S1R1V1E1R1A1E1N1V1A1F1AN1K1O1H1Q1L
d1l1f1f1de1ant1te1as1c1e1k1ev1l1cd1h1m1d1v1g1p1aed1sg1h1ia1A1C1N1P1Y1K1H1E1M1I1CR1E1S1AG1V1R1V1M1E1T1AD1R1G1S1P
LR1Q1L1V1Y1H1AL1P1S1L1P1M1V1D1F1G1QL1S1D1VA1K1Y1I1Q1V1Q1R1V1ES1I1S1D1EN1G1TR1V1te1v1c1as1q1g1k1r1ted1ec1f1s1v1rd
q1v1h1v1s1f1c1s1p1h1t1p1q1I1T1F1TR1Q1C1AR1F1Q1Q1D1L1Q1Y1V1V1V1D1E1V1G1aed1sp1k1m1k1t1l1h1L1D1E1G1D1ED1D1P1A1K1H1K1V1G
FV1G1N1W1AL1D1P1AK1M1NR1G1F1V1S1R1GS1P1NET1EL1ES1AK1G1c1ss1dl1v1q1dr1v1q1y1f1as1f1ak1ay1et1v1c1r1qd1k1f1rd1y1v1S1I
1K1M1V1AA1K1AS1NR1K1P1S1P1D1LA1Q1AV1R1N1F1S1G1K1D1D1Q1AL1D1FL1AN1LE1PA1E1AK1E1S1E1FM1Q1L1K1E1I1F1G1P1S1Q1V1E1G1Q1D1E1AS1R
Y1L1V1L1T1K1N1V1AL1Q1I1Q1T1F1F1E1G1d1q1p1e1i1f1g1s1f1k1d1q1e1y1t1q1ern1rv1k1m1et1g1k1m1l1n1l1n1q1y1e1L1W1AL1N1Q1Y1
V1H1L1G1Q1Y1V1D1L1G1L1T1R1V1K1R1V1H1P1N1F1R1V1I1E1E1K1D1V1Y1K1H1F1I1P1L1N1R1E1K1H1Y1L1d1nt1v1lek1w1k1s1ve1el1c1aw1e1k1fn
v1k1ah1f1q1r1h1k1y1s1p1d1f1g1h1s1d1CAS1V1L1Q1I1E1R1Q1P1RAL1TEL1HQ1K1V1SE1EAK1s1il1n1cat1pd1av1r1s1ays1lg1FAA
E1W1LS1Q1E1Y1F1H1R1H1NS1F1AD1FL1Q1A1H1L1T1A1der1ha1f1te1it1f1s1r1L1T1SH1D1CE1LE1SE1V1T1R1A1P1K1L1L1W1L1Q1D1T1E1Y1S1L1K
E1VR1N1C1L1N1T1AR1K1L1I1P1Q1T1F1E1D1G1R1S1A1Q1L1AS1AK1Y1S1V1NE1ink1ire1ned1r1fy1f1it1k1s1v1r1g1t1ay1v1G1H1L1W1Q1S1V
H1D1L1R1R1T1M1V1SD1V1T1R1Q1H1V1T1S1QL1F1AP1G1D1PE1L1G1E1H1R1A1ED1G1EE1A1M1E1A1S1T1s1ge1va1e1A1E1E1M1T1S1E1S1E1V1G1K1E1S
EL1G1G1S1D1V1I1L1D1T1r1l1r1sc1v1s1av1g1m1rd1q1n1e1s1ct1r1m1r1v1v1l1g1l1n1e1dd1a1cha1S1FL1R1V1S1K1R1L1S1V1L1K1K1Q1E1S1Q1F1H
LE1WL1R1e1ac1n1g1i1v1v1q1nh1N1L1S1E1N1N1V1F1S1W1K1I1D1Y1E1EL1W1V1Q1A1Y1I1T1DA1E1GL1PK1F1VD1I1F1Q1T1P1L1R1FL1A1L1HG1E1P1Q
Q1E1L1Q1CY1L1R1D1F1L1T1M1R1V1S1TE1E1L1K1F1L1Q1M1AL1W1s1ct1r1k1ka1e1s1e1e1e1v1l1p1w1h1L1AY1Q1R1F1R1S1L1Q1F1R1I1T1Y1P1Q1V1L1H
S1L1E1A1r1wn1hel1ag1cem1L1DA1F1A1M1A1C1T1E1M1L1R1N1T1L1K1P1S1Q1A1W1L1Q1V1K1n1sm1p1el1i1cs1d1eh1m1g1s1g1LA1Q1V1R1E1V1R1A1Q1W
S1R1I1F1S1T1AL1F1V1E1H1L1G1T1E1S1R1V1P1E1L1Q1L1V1T1E1H1V1L1D1k1cl1r1ens1dv1k1t1hg1fe1av1m1t1ce1ket1a1sk1t1er1f1g1p1c1s1ic
1gd1ak1dp1v1elp1cd1h1v1cl1r1cl1raw1f1ase1gm1c1p1y1c1l1tal1p1d1ef1sp1av1s1q1h1r1e1a1e1k1h1r1f1g1m1c1ns1f1vd1V1S1T1CF1KD
N1AP1E1E1E1E1S1L1L1S1L1V1Q1K1R1L1D1A1Q1R1H1CE1K1S1L1S1F1N1D1V1D1K1T1P1V1R1S1V1L1K1L1L1K1L1Y1I1Q1E1Y1L1T1L1K1K1
AF1t1ed1kt1ely1m1f1in1LED1S1I1E1K1T1S1A1Y1S1R1N1E1L1N1H1L1E1E1G1R1F1L1K1A1Y1S1P1AS1G1R1E1P1A1N1E1A1S1V1E1Y1Q1E1A1V1R1L1C1L1D1R1A1
D1F1L1S1E1P1E1G1P1E1M1A1k1e1k1q1y1l1q1q1k1f1c1ir1vend1W1H1R1V1L1V1R1K1L1S1Q1R1G1E1F1V1Q1L1S1K1P1G1R1H1Q1W1F1F1D1V1V1Q1G1L1R1D1H
F1G1Q1M1D1R1Y1V1G1E1Y1K1AL1R1DA1V1ak1av1LE1CK1FL1G1IK1T1ALK1ACK1T1P1Q1S1Q1S1AY1F1L1L1F1R1E1V1A1L1Y1S1H1N1A1S1L1H1P1E1Q1C1E1AV
SK1F1G1E1CK1L1S1P1D1S1R1F1ATS1L1V1D1NS1V1F1L1R1Ag1ps1ds1n1dgt1vt1em1ai1ha1s1a1v1l1eg1p1e1l1eg1p1e1l1k1n1a1S1p1at1ma1h1f1l
p1mp1ed1l1ag1ar1w1k1e1rv1h1w1t1c1p1n1g1h1c1s1v1g1e1c1r1p1m1e1s1c1DC1E1H1P1G1G1D1H1R1P1D1G1F1H1K1D1AD1T1P1O1H1V1G1
N1P1QR1D1V1t1cd1r1g1p1v1f1l1r1l1th1a1l1l1g1s1q1S1Q1AL1N1I1K1P1V1R1D1P1G1F1Q1H1L1K1D1E1Q1L1M1G1H1G1A1D1E1T1G1
V1H1L1R1R1L1Q1E1Q1L1S1R1L1L1N1D1T1E1L1S1T1K1E1M1N1N1E1K1E1A1V1I1S1PE1L1H1L1D1T1L1P1M1N1I1S1Q1K1R1S1M1V1K1A1I1Y1G
D1P1V1T1P1H1L1R1K1S1V1H1C1K1I1W1S1C1R1K1R1I1V1E1Y1L1Q1H1V1E1Q1R1N1K1R1V1I1L1H1F1L1Q1K1E1A1E1L1R1V1K1F1E1L1A1L1Q1D1V1L1Q1F1Q1N
V1Q1V1E1S1S1R1G1f1L1SK1H1S1D1G1L1R1L1H1N1I1T1V1F1L1S1W1N1K1L1R1S1L1E1T1N1E1I1N1L1P1K1D1Y1C1s1d1d1d1ef1e1l1l1pr1r1g1l1c1a
tal1v1s1y1l1r1H1N1E1I1V1A1E1K1L1S1K1N1S1Y1V1D1A1E1V1T1E1H1w1s1y1e1v1er1d1t1p1l1ns1c1y1q1e1e1r1v1e1f1d1e1k1r1q1Q1
V1S1R1F1L1Q1K1P1R1L1S1K1G1I1P1T1V1Y1R1H1D1N1Y1E1H1F1M1D1K1N1K1a1q1d1l1p1s1v1s1a1s1g1l1q1s1y1d1e1v1s1v1e1v1t1gl1f1st1agg
d1p1nm1l1n1y1t1q1l1q1m1g1Q1T1H1V1L1K1AL1N1R1C1L1K1H1T1AL1W1Q1L1S1A1H1K1E1S1Q1L1L1R1L1K1E1P1G1E1S1S1Y1K1AD1L1S1P1N1AK1L1S1T1F
L1n1qt1L1DA1F1L1E1L1H1E1M1I1L1K1L1N1P1Q1T1E1R1F1R1P1Q1S1L1R1D1L1S1V1M1Q1K1E1S1I1P1emas1q1f1p1e1il1asc1v1s1v1k1ta1v1l
k1wn1rem1r

```

3. The authors have followed protocol and deposited their mass spectrometry data into a public repository (PRIDE). However, it is somewhat unclear why they did this in two different deposits, what is what? - please clarify.

Answer: Indeed, the mass spectrometry data have been deposited in two distinct PRIDE projects in order to describe the two datasets in detail without word limitation. This was necessary because both experiments relied on different, complementary methods. The data for the Virotrap experiments was submitted with the identifier PXDo18345 and with the title: "Mapping of ISG15 interaction partners by Virotrap coupled to mass spectrometry", while the data for the GST pull down experiments was submitted under the identifiers PXDo18346 and with the title: "Mapping of ISG15 interaction partners by GST pull down coupled to mass spectrometry". We slightly modified the methods section to make this more clear and also indicated the PRIDE identifiers in the Data Availability statement.

4. As the authors stated, RNF213 is a ubiquitin E₃ ligase of the RING finger family. At this stage, it is not clear what its role is as a ubiquitin E₃ ligase in this process. Is there a possible interplay and/or regulation by ISG15 binding at these critical sites described by the authors?

Answer: While this paper was in review, the lab of Felix Randow (Otten et al., Nature 2021) published that RNF213 acts as a ubiquitin ligase that can directly ubiquitinate cell wall components of cytosolic bacteria. In their experiments, the authors used *Salmonella* (Gram- bacterium) and showed that RNF213 can transfer ubiquitin on the lipid A moiety of LPS. Our findings with *Listeria* (Gram+ bacterium, lacking LPS) suggest that also other bacterial cell wall components can be targeted by RNF213, something that was indicated in a recent commentary on the Randow paper (Damgaard and Pruneda, Mol Cell 2021, citing our manuscript from bioRxiv) and that we would like to address in future studies. The Randow lab showed that ubiquitination of cytosolic *Salmonella* by RNF213 initiates xenophagic clearance. Here again, differences can be expected with cytosolic *Listeria* which, in contrast to *Salmonella*, can evade xenophagy through various mechanisms. The role of ISG15 in this process is another outstanding question. Interestingly, our data suggest that the RNF213 antilisterial activity is dependent on ISG15, likely since ISGylation of RNF213 itself is required for its oligomerization on lipid droplets as well as bacteria. However, it is well possible that RNF213 counteracts *Listeria* using different mechanisms compared to *Salmonella*. Maybe the antilisterial activity of RNF213 is rather linked to its targeting of ISGylated proteins to lipid droplets, somehow restricting intracellular energy sources. Another exciting possibility that we would like to investigate in the future is whether RNF213 could act as a potential ISG15 ligase, next to its ubiquitin ligase activity. Maybe RNF213's naming RING domain, which was shown to be obsolete for ubiquitin ligase activity, could instead transfer ISG15? In any case, the very strong *in vivo* phenotype that we observe in the RNF213 KO animals indicates a key role of RNF213 to restrict *Listeria*. Along with other experiments, we are planning to cross these animals with ISG15 KO animals to better dissect the role of both proteins in the defense against *Listeria* and other pathogens.

REVIEWERS' COMMENTS

Reviewer #1 (Remarks to the Author):

My concerns by the reviewers have been adequately addressed. The changes made have strengthened the manuscript.

Reviewer #2 (Remarks to the Author):

The authors have done a great job addressing my comments. This is an exciting paper that opens up many new questions about cell intrinsic immunity.